# Do Residual Neural Networks discretize Neural Ordinary Differential Equations?

**Michael E. Sander**
ENS, CNRS
Paris, France
michael.sander@ens.fr

**Pierre Ablin**
Université Paris-Dauphine, CNRS
Paris, France
pierreablin@gmail.com

**Gabriel Peyré**
ENS, CNRS
Paris, France
gabriel.peyre@ens.fr

## Abstract

Neural Ordinary Differential Equations (Neural ODEs) are the continuous analog of Residual Neural Networks (ResNets). We investigate whether the discrete dynamics defined by a ResNet are close to the continuous one of a Neural ODE. We first quantify the distance between the ResNet's hidden state trajectory and the solution of its corresponding Neural ODE. Our bound is tight and, on the negative side, does not go to $0$ with depth $N$ if the residual functions are not smooth with depth. On the positive side, we show that this smoothness is preserved by gradient descent for a ResNet with linear residual functions and small enough initial loss. It ensures an implicit regularization towards a limit Neural ODE at rate $\frac{1}{N}$, uniformly with depth and optimization time. As a byproduct of our analysis, we consider the use of a memory-free discrete adjoint method to train a ResNet by recovering the activations on the fly through a backward pass of the network, and show that this method theoretically succeeds at large depth if the residual functions are Lipschitz with the input. We then show that Heun's method, a second order ODE integration scheme, allows for better gradient estimation with the adjoint method when the residual functions are smooth with depth. We experimentally validate that our adjoint method succeeds at large depth, and that Heun's method needs fewer layers to succeed. We finally use the adjoint method successfully for fine-tuning very deep ResNets without memory consumption in the residual layers.

## 1 Introduction

**Problem setup.** Residual Neural Networks (ResNets) [20, 21] keep on outperforming state of the art in computer vision [46, 6], and more generally skip connections are widely used in a various range of applications [42, 14]. A ResNet of depth $N$ iterates, starting from $x_0 \in \mathbb{R}^d$, $x_{n+1} = x_n + f(x_n, \theta_n^N)$ and outputs a final value $x_N \in \mathbb{R}^d$ where $f$ is a neural network called residual function. In this work, we consider a simple modification of this forward rule by letting explicitly the residual mapping depend on the depth of the network:

$$x_{n+1} = x_n + \frac{1}{N} f(x_n, \theta_n^N). \tag{1}$$

On the other hand, a Neural ODE [8] uses a neural network $\varphi_\Theta(x, s)$, that takes time $s$ into account, to parameterise a vector field [24] in a differential equation, as follows,

$$\frac{\mathrm{d}x}{\mathrm{d}s} = \varphi_\Theta(x(s), s) \quad \text{with} \quad x(0) = x_0, \tag{2}$$

and outputs a final value $x(1) \in \mathbb{R}^d$, the solution of Eq.(2). The Neural ODE framework enables learning without storing activations (the $x_n$'s) using the adjoint state method, hence significantly reducing the memory usage for backpropagation that can be a bottleneck during training [43, 34, 48, 16].

36th Conference on Neural Information Processing Systems (NeurIPS 2022).

Neural ODEs also provide a theoretical framework to study deep learning models from the continuous viewpoint, using the arsenal of ODE theory [40, 25, 41]. Importantly, they can also be seen as the continuous analog of ResNets. Indeed, consider for $N$ an integer, the Euler scheme for solving Eq. (2) with time step $\frac{1}{N}$ starting from $x_0$ and iterating $x_{n+1} = x_n + \frac{1}{N}\varphi_\Theta(x_n, \frac{n}{N})$. Under mild assumptions on $\varphi_\Theta$, this scheme is known to converge to the true solution of Eq. (2) as $N$ goes to $+\infty$. Also, if $\Theta = (\theta_n^N)_{i \in [N-1]}$ and $\varphi_\Theta(., \frac{n}{N}) = f(., \theta_n^N)$, then the ResNet equation Eq. (1) corresponds to a Euler discretization with time step $\frac{1}{N}$ of Eq. (2). However, for a given ResNet with fixed depth $N$ and weights, the activations in Eq. (1) can be far from the solution of Eq. (2). This is illustrated in Figure 1 where we show that a deep ResNet can easily break the topology of the input space, which is impossible for a

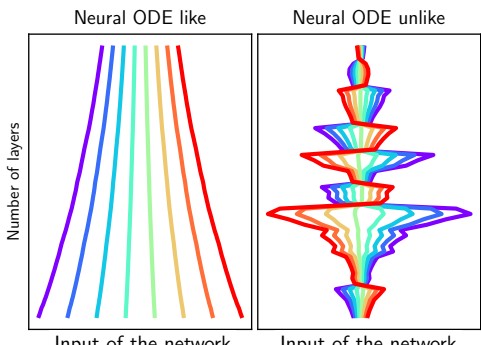

Figure 1: **Trajectory of ResNets with 300 layers.** Left: we learn $x \to \frac{x^2}{2}$, trajectories are smooth and do not intersect. Right: we learn $x \to \frac{-x^2}{2}$, trajectories are not smooth and intersect.

Neural ODE. In this paper, we study the link between ResNets and Neural ODEs. We make the following contributions:

- In Section 3, we propose a framework to define a set of associated Neural ODEs for a given ResNet. We control the error between the discrete and the continuous trajectory. We show that without additional assumptions on the smoothness with depth of the residual functions, this error does not go to 0 as $N \to \infty$ (Prop. 1). However, we show that under some assumptions on the weight initialization, the trained parameters of a deep linear ResNet uniformly (with respect to both depth and training time) approach a Lipschitz function as the depth $N$ of the network goes to infinity, at speed $\frac{1}{N}$ (Prop. 2 and Th. 1). This result highlights an implicit regularization towards a limit Neural ODE.

- In Section 4, we investigate a simple technique to train ResNets without storing activations. Inspired by the adjoint method, we propose to recover the approximated activations during the backward pass by using a reverse-time Euler scheme. We control the error for recovering the activations and gradients with this method. We show that if the residuals of the ResNet are bounded and Lipschitz continuous, with constants independent of $N$, then this error scales in $O(\frac{1}{N})$ (Prop. 3). Hence, the adjoint method needs a large number of layers to lead to correct gradients (Prop. 4). We then consider a smoothness-dependent reconstruction with Heun's method to bound the error between the true and approximated gradient by a term that depends on $\frac{1}{N}$ times the smoothness in depth of the residual functions, hence guaranteeing a better approximation when successive weights are close one to another (Prop. 5 and 6).

- In Section 5, on the experimental side, we show that the adjoint method fails when training a ResNet 101 on ImageNet. Nevertheless, we empirically show that very deep ResNets pretrained with tied weights (constant weights: $\theta_n^N = \theta \; \forall n$) can be refined -using our adjoint method- on CIFAR-10 and ImageNet by untying their weights, leading to a better test accuracy. Last, but not least, we show using a ResNet architecture with heavy downsampling in the first layer that our adjoint method succeeds at large depth and that Heun's method leads to a better behaved training, hence confirming our theoretical results.

## 2  Background and related work

**Neural ODEs.**  Neural ODEs are a class of implicit deep learning models defined by an ODE where a neural network parameterises the vector field [44, 8, 40, 39, 45, 29, 36, 24]. Given an input $x_0$, the output of the model is the solution of the ODE (2) at time 1. From a theoretical viewpoint, the expression capabilities of Neural ODEs have been investigated in [12, 41, 25] and the Neural ODE framework has been used to better understand the dynamics of more general architectures that include residual connections such as Transformers [38, 27]. Experimentaly, Neural ODEs have been successful in a various range of applications, among which physical modelling [18, 11] and generative modeling [8, 17]. However, there are many areas where Neural ODEs have failed to replace ResNets, for instance for building computer vision classification models. Neural ODEs fail to compete with

ResNets on ImageNet, and to the best of our knowledge, previous works using Neural ODEs on ImageNet consider weight-tied architectures and only achieves the same accuracy as a ResNet18 [50]. It has also been shown that Neural ODEs sometimes do not admit a continuous interpretation at all [32].

**Implicit Regularization of ResNets towards ODEs.** Recent works have studied the link between ResNets and Neural ODEs. In [9], the authors carry experiments to better understand the scaling behavior of weights in ResNets as a function of the depth. They show that under the assumption that there exists a scaling limit $\theta(s) = N^\beta \lim \theta^N_{\lfloor Ns \rfloor}$ for the weights of the ResNets (with $0 < \beta < 1$) and if the scale of the ResNet is $\frac{1}{N^\alpha}$ with $0 < \alpha < 1$ and $\alpha + \beta = 1$, then the hidden state of the ResNet converges to a solution of a linear ODE. In this paper, we are interested in the case where $\alpha = 1$, which seems more natural since it is the scaling that appears in Euler's method with step $\frac{1}{N}$. In addition, we do not assume the existence of a scaling limit $\theta(s) = \lim \theta^N_{\lfloor Ns \rfloor}$. In subsection 3.2, we demonstrate the existence of this scaling limit in the linear setting, under some assumptions. The recent work [10] shows results regarding linear convergence of gradient descent in ResNets and prove the existence of an $\frac{1}{2}$-Hölder continuous scaling limit as $N \to \infty$ with a scaling factor for the residuals in $\frac{1}{\sqrt{N}}$ which is different from ours ($\frac{1}{N}$). In contrast, we show that our limit function is Lipschitz continuous, which is a stronger regularity. This is to be linked with the recent work of [31], where the authors show that whereas the $\frac{1}{\sqrt{N}}$ scaling corresponds to the proper one for standard i.i.d. initializations, $\frac{1}{N}$ is the proper scaling for smooth initialization to obtain non-trivial behaviour (other choices lead to explosion or to identity [31]). More generally, recent works have proved the convergence of gradient descent training of ResNet when the initial loss is small enough. This include ResNet with finite width but arbitrary large depth [15, 26] and ResNet with both infinite width and depth [28, 3]. These convergence proofs leverage an implicit bias toward weights with small amplitudes. They however leave open the question of convergence of individual weights as depth increases, which we tackle in this work in the linear case. This requires showing an extra bias toward weights with small variations across depth.

**Memory bottleneck in ResNets.** Training deep learning models involve graphics processing units (GPUs) where memory is a practical bottleneck [43, 34, 48]. Indeed, backpropagation requires to store activations at each layer during the forward pass. Since samples are processed using mini batches, this storage can be important. For instance, with batches of size 128, the memory needed to compute gradients for a ResNet 152 on ImageNet is about 22 GiB. Note that the memory needed to store the parameters of the model is only 220 MiB, which is negligible compared to the memory needed to store the activations. Thus, designing deep invertible architectures where one can recover the activations on the fly during the backpropagation iterations has been an active field in recent years [16, 37, 23]. In this work, we propose to approximate activations using a reverse-time Euler scheme, as we detail in the next subsection.

**Adjoint Method.** Consider a loss function $L(x_N)$ for the ResNet (1). The backpropagation equations [5] are

$$\nabla_{\theta^N_{n-1}} L = \frac{1}{N}[\partial_\theta f(x_{n-1}, \theta^N_{n-1})]^\top \nabla_{x_n} L, \quad \nabla_{x_{n-1}} L = [I + \frac{1}{N}\partial_x f(x_{n-1}, \theta^N_{n-1})]^\top \nabla_{x_n} L. \quad (3)$$

Now, consider a loss function $L(x(1))$ for the Neural ODE (2). The adjoint state method [35, 8] gives

$$\nabla_\Theta L = -\int_T^0 \partial_\Theta [\varphi_\Theta(x(s), s)]^\top \nabla_{x(s)} L \, ds, \quad -\dot{\nabla}_{x(s)} L(s) = [\partial_x \varphi_\Theta(x(s), s)]^\top \nabla_{x(s)} L. \quad (4)$$

Note that if $\Theta = (\theta^N_n)_{n \in [N-1]}$ and $\varphi_\Theta(., \frac{n}{N}) = f(., \theta^N_n)$, then Eq. (3) corresponds to a Euler discretization with time step $\frac{1}{N}$ of Eq. (4). The key advantage of using Eq. (4) is that one can recover $x(s)$ on the fly by solving the Neural ODE (2) backward in time starting from $x(1)$. This strategy avoids storing the forward trajectory $(x(s))_{s \in [0,1]}$ and leads to a $O(1)$ memory footprint [8]. In this work, we propose to use a discrete adjoint method by using a reverse-time Euler scheme for approximately recovering the activations in a ResNet (Section 4). Contrarily to other models such as RevNets [16] (architecture change) or Momentum ResNets [37] (forward rule modification) which rely on an exactly invertible forward rule, the proposed method requires no change at all in the network, but gives approximate gradients.

**Notations.** For $k \in \mathbb{N}$, $\mathcal{C}^k$ is the set of functions $f : \mathbb{R}^d \to \mathbb{R}^d$ $k$ times differentiable with continuous $k^{th}$ differential. If $f \in \mathcal{C}^1$, $\partial_x f(x)[y]$ is the differential of $f$ at $x$ evaluated in $y$. For $K \subset \mathbb{R}^d$ compact, $\|.\|$ a norm and $f$ a continuous function on $\mathbb{R}^d$, we denote $\|f\|_\infty^K = \sup_{x \in K} \|f(x)\|$.

## 3 ResNets as discretization of Neural ODEs

In this section we first show that without further assumptions, the distance between the discrete trajectory and the solution of associated ODEs can be constant with respect to the depth of the network if the residual functions lack smoothness with depth. We then present a positive result by studying the linear case where we show that, under some hypothesis (small loss initialization and initial smoothness with depth), the ResNet converges to a Neural ODE as the number of layers goes to infinity. We show that this convergence is uniform with depth and optimization time.

### 3.1 Distance to an ODE

We first define associated Neural ODEs for a given ResNet.

**Definition 1.** *We say that a neural network $\varphi_\Theta : \mathbb{R}^d \times \mathbb{R} \to \mathbb{R}^d$ smoothly interpolates the ResNet Eq.* (1)*. if $\varphi_\Theta$ is smooth and $\forall n \in \{0, ..., N-1\}$, $\varphi_\Theta(., \frac{n}{N}) = f(., \theta_n^N)$.*

Note that we omit the dependency of $\Theta$ in $N$ to simplify notations. For example, for a given ResNet, there are two natural ways to interpolate it with a Neural ODE, either by interpolating the *residuals*, or by interpolating the *weights*. Indeed, one can interpolate the residuals with $\varphi_\Theta(\cdot, s) = (n + 1 - Ns)f(., \theta_n^N) + (Ns - n)f(\cdot, \theta_{n+1}^N)$ when $s \in [\frac{n}{N}, \frac{n+1}{N}]$, or interpolate the weights with $\varphi_\Theta(\cdot, s) = f(\cdot, (n+1-Ns)\theta_n^N + (Ns-n)\theta_{n+1}^N)$ for $s \in [\frac{n}{N}, \frac{n+1}{N}]$. If $\theta_n^N = \theta^N$ does not depend on $n$, then both interpolations are identical and one can simply consider $\varphi_\Theta(x, s) = f(x, \theta^N), \forall (x, s)$.

We now consider any smooth interpolation $\varphi_\Theta$ for the ResNet (1) and a Euler scheme for the Neural ODE (2) with time step $\frac{1}{N}$.

**Proposition 1** (Approximation error). *We suppose that $\varphi_\Theta$ is $\mathcal{C}^1$, and $L$-Lipschitz with respect to $x$, uniformly in $s$. Note that this implies that the solution of Eq.* (2) *is well defined, unique, $\mathcal{C}^2$, and that the trajectory is included in some compact $K \subset \mathbb{R}^d$. Denote $C_N := \|\partial_s \varphi_\Theta + \partial_x \varphi_\Theta[\varphi_\Theta]\|_\infty^{K \times [0,1]}$. Then one has for all $n$: $\|x_n - x(\frac{n}{N})\| \leq \frac{e^L - 1}{2NL} C_N$ if $L > 0$ and $\|x_n - x(\frac{n}{N})\| \leq \frac{C_N}{2N}$ if $L = 0$.*

For a full proof, see appendix A.1. Note that this result extends Theorem 3.2 from [49] to the non-autonomous case: our bound depends on $\partial_s \varphi_\Theta$. Finally, our bound is tight. Indeed, for $\varphi_\Theta(x, s) = as + b$ for $a, b \in \mathbb{R}^d$, we get $\partial_s \varphi_\Theta + \partial_x \varphi_\Theta[\varphi_\Theta] = a$, $L = 0$ and $\|x(1) - x_N\| = \frac{\|a\|}{2N}$.

**Implication.** The tightness of our bound shows that closeness to the ODE solution is not guaranteed, because we do not know whether $C_N/N \to 0$. Indeed, consider first the residual interpolation $\varphi_\Theta(x, s) = \left((n + 1 - Ns)f(x, \theta_n^N) + (Ns - n)f(x, \theta_{n+1}^N)\right)\mathbb{1}_{s \in [\frac{n}{N}, \frac{n+1}{N}]}$ and the simple case where $\partial_x \varphi_\Theta[\varphi_\Theta] = 0$. We get $\partial_s \varphi_\Theta(x, s) = N(f(x, \theta_{n+1}^N) - f(x, \theta_n^N))\mathbb{1}_{s \in [\frac{n}{N}, \frac{n+1}{N}]}$, which corresponds to the discrete derivative. It means that although there is a $\frac{1}{N}$ factor in our bound, the time derivative term – without further regularity with depth of the weights, which is at the heart of subsection 3.2 – usually scales with $N$: $\frac{C_N}{N} = O(1)$. As a first example, consider the simple case where $f(x, \theta_n^N) = n$. This gives $x_N = x_0 + \frac{(N-1)}{2}$ while the integration of the Neural ODE (2) leads to $x(1) = x_0 + \frac{N}{2}$ because $\varphi_\Theta(x, s) = Ns$, so the $\|x_N - x(1)\| = \frac{1}{2}$ is not small. Intuitively, this shows that weights cannot scale with depth when using the residual interpolation. Now, consider the weight interpolation, $\theta_n^N = (-1)^n$ and suppose $f$ is written as $f(x, \theta) = \theta^2$. This gives $\varphi_\Theta(., s) = (2Ns - (2n + 1))^2$ when $s \in [\frac{n}{N}, \frac{n+1}{N}]$. Integrating, we get $x(1) = \frac{1}{3}$ while the output of the ResNet is $x_N = 1$. Hence $\|x_N - x(1)\| = \frac{2}{3}$ is also not small, even though the weights are bounded. Thus, one needs additional regularity assumptions on the weights of the ResNet to obtain a Neural ODE in the large depth limit. Intuitively if the weights are initialized close from one another and they are updated using gradient descent, they should stay close from one another during training, since the gradients in two consecutive layers will be similar, as highlighted in Eq. (3). Indeed, we see that if $x_n$ and $x_{n+1}$ are close, then $\nabla_{x_n} L$ and $\nabla_{x_{n+1}} L$ are close, and then if $\theta_n^N$ and $\theta_{n+1}^N$ are close, $\nabla_{\theta_n^N} L$ and $\nabla_{\theta_{n+1}^N} L$ are also close. In subsection 3.2, we formalize this intuition and present a

positive result for ResNets with linear residual functions. More precisely, we show that with proper initialization, the difference between two successive parameters is in $\frac{1}{N}$ during the entire training. Furthermore, we show that the weights of the network converge to a smooth function, hence defining a limit Neural ODE.

## 3.2 Linear Case

As a further step towards a theoretical understanding of the connections between ResNets and Neural ODEs we investigate the linear setting, where the residual functions are written $f(x, \theta) = \theta x$ for any $\theta \in \mathbb{R}^{d \times d}$. It corresponds to a deep matrix factorization problem [51, 4, 2, 1]. As opposed to these previous works, we study the infinite depth limit of these linear ResNets with a focus on the learned weights. We show that, if the weights are initialized close one to another, then at any training time, the weights stay close one to another (Prop. 2) and importantly, they converge to a smooth function of the continuous depth $s$ as $N \to \infty$ (Th. 1). All the proofs are available in appendix A.

**Setting.** Given a training set $(x_k, y_k)_{k \in [n]}$ in $\mathbb{R}^d$, we solve the regression problem of mapping $x_k$ to $y_k$ with a linear ResNet, *i.e.* $f(x, \theta) = \theta x$, of depth $N$ and parameters $(\theta_1^N, \ldots, \theta_N^N)$. The ResNet therefore maps $x_k$ to $\Pi^N x_k$ where $\Pi^N := \prod_{n=1}^N (I_d + \frac{\theta_n^N}{N}) = (I_d + \frac{\theta_N^N}{N}) \cdots (I_d + \frac{\theta_1^N}{N})$. It is trained by minimizing the average errors $\|\Pi^N x_k - y_k\|_2^2$, which is equivalent to the deep matrix factorization problem:

$$\text{argmin}_{(\theta_n^N)_{i \in [N-1]}} L(\theta_1^N, \ldots, \theta_N^N) := \|\Pi^N - B\|_\Sigma^2, \tag{5}$$

where $\|A\|_\Sigma^2 = \text{Tr}(A\Sigma A^T)$, $\Sigma$ is the empirical covariance matrix of the data: $\Sigma := \frac{1}{n} \sum_{k=1}^n x_k x_k^\top$, and $B := \frac{1}{n} \sum_{k=1}^n y_k x_k^\top \Sigma^{-1}$. As is standard, we suppose that $\Sigma$ is non degenerated. We denote by $M > 0$ (resp. $m > 0$) its largest (resp. smallest) eigenvalue.

**Gradient.** We denote $\Pi_{:n}^N := (I_d + \frac{\theta_N^N}{N}) \cdots (I_d + \frac{\theta_{n+1}^N}{N})$ and $\Pi_{n:}^N := (I_d + \frac{\theta_{n-1}^N}{N}) \cdots (I_d + \frac{\theta_1^N}{N})$ and write the gradient $\nabla_n^N(t) = \nabla_{\theta_n^N} L(\theta_1^N(t), ..\theta_n^N(t), .., \theta_N^N(t))$. The chain rule gives $N \nabla_n^N = \Pi_{:n}^{N\top}(\Pi^N - B)\Sigma \Pi_{n:}^{N\top}$. Intuitively, as $N$ goes to $+\infty$, the products $\Pi^N$, $\Pi_{:n}^N$ and $\Pi_{n:}^N$ should converge to some limit, hence we see that $N \nabla_n^N$ scale as 1. Therefore, we train $\theta_n^N$ by the rescaled gradient flow $\frac{d\theta_n^N}{dt}(t) = -N \nabla_n^N(t)$ to minimize $L$ and denote $\ell^N(t) = L(\theta_1^N(t), \ldots, \theta_N^N(t))$.

**Two continuous variables involved.** Our results involve two continuous variables: $s \in [0, 1]$ is the depth of the limit network and corresponds to the time variable in the Neural ODE, whereas $t \in \mathbb{R}_+$ is the gradient flow time variable. As is standard in the analysis of convergence of gradient descent for linear networks, we consider the following assumption:

**Assumption 1.** *Suppose that at initialisation one has $\sqrt{\ell^N(0)} < \frac{m}{4\sqrt{2M}e^3}$ and $\|\theta_n^N(0)\| \leq \frac{1}{4}$.*

Assumption 1 is the classical assumption in the literature [51, 3] to prove linear convergence of our loss and that the $\theta_n^N(t)$'s stay bounded with $t$. Note that this bounded norm assumption implies that $\frac{1}{N}\theta_n(0) = O(\frac{1}{N})$. This is in contrast with classical initialization scales in the *feedforward* case where the initialization only depends on width [19]. However this initialization scale is coherent with those of *ResNets* for which the scale has to depend on depth [47, 31]. In addition, the experimental findings in [9] suggest that the weights in ResNets scale in $\frac{1}{N^\beta}$ with $\beta > 0$.

We now prove an implicit regularization result showing that if at initialization, in addition to assumption 1, the weights are close from one another ($O(\frac{1}{N})$), they will stay at distance $O(\frac{1}{N})$: the discrete derivative stay in $O(\frac{1}{N})$, which is a central result to consider the infinite depth limit in our Th. 1.

**Proposition 2** (Smoothness in depth of the weights). *Suppose assumption 1. Suppose that there exists $C_0 > 0$ independent of $n$ and $N$ such that $\|\theta_{n+1}^N(0) - \theta_n^N(0)\| \leq \frac{C_0}{N}$. Then, $\forall t \in \mathbb{R}_+$, $\|\theta_n^N(t)\| < \frac{1}{2}$, and $\theta_n^N(t)$ admits a limit $\psi_n^N$ as $t \to +\infty$. Moreover, there exists $C > 0$ such that $\forall t \in \mathbb{R}_+$, $\|\theta_{n+1}^N(t) - \theta_n^N(t)\| \leq \frac{C}{N}$.*

For a full proof, see appendix A.2. The inequality $\|\theta_{n+1}^N(t) - \theta_n^N(t)\| \leq \frac{C}{N}$ corresponds to a discrete Lipschitz property in depth. Indeed, for $s \in [0, 1]$ and $t \in \mathbb{R}_+$, let $\psi_N(s, t) = \theta_{\lfloor Ns \rfloor}^N(t)$. Then

our result gives $\|\psi_N(\frac{n+1}{N}, t) - \psi_N(\frac{n}{N}, t)\| \leq \frac{C}{N}$ which implies that $\|\psi_N(s_1, t) - \psi_N(s_2, t)\| \leq C|s_1 - s_2| + \frac{C}{N}$. We now turn to the infinite depth limit $N \to \infty$. Th. 1 shows that there exists a limit function $\psi$ such that $\psi_N$ converges uniformly to $\psi$ in depth $s$ and optimization time $t$. Furthermore, this limit is Lipschitz continuous in $(s, t)$. In addition, we show that the ResNet $\Pi^N$ converges to the limit Neural ODE defined by $\psi$ that is preserved along the optimization flow, exhibiting an implicit regularization property of deep linear ResNets towards Neural ODEs.

**Theorem 1** (Existence of a limit map). *Suppose assumption 1, $\|\theta_{n+1}^N(0) - \theta_n^N(0)\| \leq \frac{C_0}{N}$ for some $C_0 > 0$ and that there exists a function $\psi_{\mathrm{init}} : [0, 1] \to \mathbb{R}^{d \times d}$ such that $\psi_N(s, 0) \to \psi_{\mathrm{init}}(s)$ in $\|.\|_\infty$ uniformly in $s$ as $N \to \infty$, at speed $\frac{1}{N}$. Then the sequence $(\psi_N)_{N \in \mathbb{N}}$ uniformly converges (in $\|.\|_\infty$ w.r.t $(s, t)$) to a limit $\psi$ Lipschitz continuous in $(s, t)$ and $\|\psi - \psi_N\|_\infty = O(\frac{1}{N})$. Furthermore, $\Pi^N$ uniformly converges as $N \to \infty$ to the mapping $x_0 \to x_1$ where $x_1$ is the solution at time $1$ of the Neural ODE $\frac{\mathrm{d}x}{\mathrm{d}s}(s) = \psi(s, t)x(s)$ with initial condition $x_0$.*

We illustrate Th. 1 in Figure 2. The assumption on the existence of $\psi_{\mathrm{init}}$ ensures a convergence at speed $\frac{1}{N}$ to a Neural ODE at optimization time 0. Note that for instance, the constant initialization $\theta_n^N(0) = \theta_0 \in \mathbb{R}^{d \times d}$ satisfies this hypothesis. In order to prove Th. 1, for which a full proof is presented in appendix A.5, we first present a useful lemma: the weights of the network have at least one accumulation point.

**Lemma 1** (Existence of limit functions). *For $s \in [0, 1]$ and $t \in \mathbb{R}_+$, let $\psi_N(s, t) = \theta_{\lfloor Ns \rfloor}^N(t)$. Under the assumptions of Prop. 2, there exists a subsequence $\psi_{\sigma(N)}$ and $\psi_\sigma : [0, 1] \times \mathbb{R}_+ \to \mathbb{R}^{d \times d}$ Lipschitz continuous with respect to both parameters $s$ and $t$ such that $\psi_{\sigma(N)} \to \psi_\sigma$ uniformly (in $\|.\|_\infty$ w.r.t $(s, t)$).*

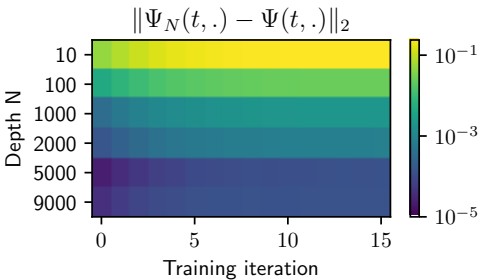

Figure 2: $L_2$ norm $\|\Psi_N(t, .) - \Psi(t, .)\|_2$ (w.r.t depth $s$) for different training iterations $t$ (horizontal axis) and different depth $N$ (vertical axis). As predicted by Th. 1, this distance goes to 0 as $N \to +\infty$.

Lemma 1 is proved in appendix A.3, and gives us the existence of a Lipschitz continuous accumulation point, but not the uniqueness nor the convergence speed. For the uniqueness, we show in appendix A.5 that, under the assumptions of Th. 1, one has that any accumulation point of $\psi_\sigma$ satisfies the limit Neural ODE

$$\partial_t \psi_\sigma(., t) = F(\psi_\sigma(., t)), \quad \psi_\sigma(., 0) = \psi_{\mathrm{init}}(., 0),$$

and show that $F$ satisfies the hypothesis of the Picard–Lindelöf theorem, hence showing the uniqueness of $\psi$. We finally show that, as intuitively expected, trajectories of the weights of our linear ResNets of depth $N$ and $2N$ remain close one to each other. This gives the convergence speed in Th. 1. See appendix A.4 for a proof.

**Lemma 2** (Closeness of trajectories). *Suppose asumption 1, $\|\theta_{n+1}^N(0) - \theta_n^N(0)\| \leq \frac{C_0}{N}$ for some $C_0 > 0$ and that $\|\theta_n^N(0) - \theta_{2n}^{2N}(0)\| = O(\frac{1}{N})$. Then $\forall t \in \mathbb{R}_+$, $\|\theta_n^N(t) - \theta_{2n}^{2N}(t)\| = O(\frac{1}{N})$.*

## 4 Adjoint Method in Residual Networks

In this section, we focus on a particularly useful feature of Neural ODEs and its applicability to ResNets: their memory free backpropagation thanks to the adjoint method. We consider a ResNet (1) and try to invert it using reverse mode Euler discretization of the Neural ODE (2) when $\varphi_\Theta$ is any smooth interpolation of the ResNet. This corresponds to defining $\tilde{x}_N = x_N$ and iterate for $n \in \{N - 1, \ldots 0\}$:

$$\tilde{x}_n = \tilde{x}_{n+1} - \frac{1}{N} f(\tilde{x}_{n+1}, \theta_n^N). \tag{6}$$

We then use the approximated activations $(\tilde{x}_n)_{n \in [N]}$ as a proxy for the true activations $(x_n)_{n \in [N]}$ to compute gradients without storing the activations:

$$\tilde{\nabla}_{\theta_{n-1}^N} L = \frac{1}{N} [\partial_\theta f(\tilde{x}_{n-1}, \theta_{n-1}^N)]^\top \nabla_{\tilde{x}_n} L, \quad \nabla_{\tilde{x}_{n-1}} L = [I + \frac{1}{N} \partial_x f(\tilde{x}_{n-1}, \theta_{n-1}^N)]^\top \nabla_{\tilde{x}_n} L. \tag{7}$$

The approximate recovery of the activations in Eq. (6) is implementable for *any* ResNet: there is no need for particular architecture or forward rule modification. The drawback is that the recovery is

only approximate. We devote the remainder of the section to the study of the corresponding errors and to error reduction using second order Heun's method. We first show that, if $f(., \theta_n^N)$ and its derivative are bounded by a constant independent of $N$, then the error for reconstructing the activations in the backward scheme (6) is $O(\frac{1}{N})$. Proofs of the theoretical results are in appendix A.

**Error for reconstructing activations.** We consider the following assumption:

**Assumption 2.** *There exists constants $C_f$ and $L_f$ such that $\forall N \in \mathbb{N}$, $\forall n \in [N-1]$, $\|f(., \theta_n^N)\|_\infty \leq C_f$ and $\|\partial_x[f(., \theta_n^N)]\|_\infty \leq L_f$.*

Then the error made by reconstructing the activations is in $O(\frac{1}{N})$.

**Proposition 3** (Reconstruction error). *With assumption 2, one has $\|x_n - \tilde{x}_n\| \leq \frac{(e^{L_f}-1)C_f}{N} + O(\frac{1}{N^2})$.*

Prop. 3 shows a slow convergence of the error for recovering activations. This bound does not depend on the discrete derivative $f(., \theta_{n+1}^N) - f(., \theta_n^N)$, contrarily to the errors between the ResNet activations and the trajectory of the interpolating Neural ODE in Prop 1. In summary, even though regularity in depth is necessary to imply closeness to a Neural ODE, it is not necessary to recover activations, and neither gradients, as we now show.

**Error in gradients when using the adjoint method.** We use the result obtained in Prop. 3 to derive a bound in $O(\frac{1}{N^2})$ on the error made for computing gradients using formulas (7).

**Proposition 4** (Gradient error). *Suppose assumption 2. Suppose in addition that $\partial_x f(., \theta)$ admits a Lipschitz constant $L_{df}$, $\partial_\theta f(., \theta)$ admits a Lipschitz constant $\Delta$, and an upper bound $\Omega$, all of which are independent of $\theta$. Then one has $\|\tilde{\nabla}_{\theta_n^N} L - \nabla_{\theta_n^N} L\| = O(\frac{1}{N^2})$.*

For a proof, see appendix A.7, where we give the dependency of our upper bound as a function of $\Delta, L_f, C_f, \Omega$ and $L_{df}$.

**Smoothness-dependent reconstruction with Heun's method.** The bounds in Prop. 3 and 4 do not depend on the smoothness with respect to the weights of the $f(., \theta_n^N)$. Only the magnitude of the residuals plays a role in the correct recovery of the activations and estimation of the gradient. Hence, there is no apparent benefit of having such a network behave like a Neural ODE. We now turn to Heun's method, a second order integration scheme, and show that in this case smoothness in depth of the network improves activation recovery. A HeunNet [30] of depth $N$ with parameters $\theta_1^N, \ldots, \theta_N^N$ iterates for $n = 0, \ldots, N-1$:

$$y_n = x_n + \frac{1}{N}f(x_n, \theta_n^N) \quad \text{and} \quad x_{n+1} = x_n + \frac{1}{2N}(f(x_n, \theta_n^N) + f(y_n, \theta_{n+1}^N)). \quad (8)$$

These forward iterations can once again be approximately reversed by doing for $n = N-1, \ldots, 0$:

$$\tilde{y}_n = \tilde{x}_{n+1} - \frac{1}{N}f(\tilde{x}_{n+1}, \theta_{n+1}^N) \quad \text{and} \quad \tilde{x}_n = \tilde{x}_{n+1} - \frac{1}{2N}(f(\tilde{x}_{n+1}, \theta_{n+1}^N) + f(\tilde{y}_n, \theta_n^N)), \quad (9)$$

which also enables approximated backpropagation without storing activations. When discretizing an ODE, Heun's method has a better $O(\frac{1}{N^2})$ error, hence we expect a better recovery than in Prop. 3. Indeed, we have:

**Proposition 5** (Reconstruction error - Heun's method). *Assume assumption 2. Denote by $L'_f$ the Lipschitz constant of $x \mapsto \frac{1}{2}(f(x, \theta_{n+1}^N) + f(x - \frac{1}{N}f(x, \theta_{n+1}^N), \theta_n^N))$, by $L_\theta$ the Lispchitz constant of $\theta \mapsto f(\cdot, \theta)$ and by $L'_\theta$ that of $\theta \mapsto \partial_x f(., \theta)$. Let $C'_f = \frac{1}{4}L'_\theta L_\theta$. Finally, define $\Delta_\theta^N := \max_n \|\theta_{n+1}^N - \theta_n^N\|^2$. Using Heun's method, we have: $\|x_n - \tilde{x}_n\| \leq \frac{(e^{L'_f}-1)C'_f}{L'_f N} \times \Delta_\theta^N + O(\frac{1}{N^2})$.*

This bound is very similar to that in proposition 3, with an additional factor $\Delta_\theta^N$. Hence, we see that under the condition that $\Delta_\theta^N = O(\frac{1}{N})$, the reconstruction error $\|x_n - \tilde{x}_n\|$ is in $O(\frac{1}{N^2})$. In the linear case, we have proven under some hypothesis in Prop. 2 that such a condition on $\Delta_\theta^N$ holds during training. Consequently, the smoothness of the weights of a HeunNet in turns helps it recover the activations, while it is not true for a ResNet. This provides better guarantees on the error on gradients:

**Proposition 6** (Gradient error - Heun's method). *Suppose assumption 2. Suppose in addition that $\partial_x f(.,\theta)$ admits Lipschitz constant, $\partial_\theta f(.,\theta)$ admits a Lipschitz constant and an upper bound, all of which are independent of $\theta$. Then one has $\|\tilde{\nabla}_{\theta_n^N} L - \nabla_{\theta_n^N} L\| = O(\frac{\Delta_\theta^N}{N^2} + \frac{1}{N^3})$.*

Just like with activation, we see that Heun's method allows for a better gradient estimation when the weights are smooth with depth. Equivalently, for a fixed depth, this proposition indicates that HeunNets have a better estimation of the gradient with the adjoint method than ResNets which ultimately leads to better training and overall better performances by such memory-free model.

## 5 Experiments

We now present experiments to investigate the applicability of the results presented in this paper. We use Pytorch [33] and Nvidia Tesla V100 GPUs. Our code is available on GitHub. All the experimental details are given in appendix B, and we provide a recap on ResNet architectures in appendix C.

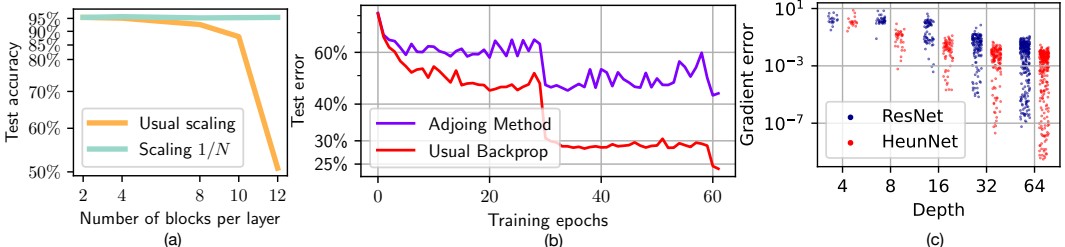

Figure 3: (a) **Test accuracy on CIFAR-10** as a function of the number of blocks in each layer of the ResNet. Within each layer, weights are tied (3 runs). (b) **Failure** of the adjoint method with a ResNet-101 on ImageNet (the approximated gradients are only used in the third layer of the network, that contains 23 blocks). (c) **Relative error** between the approximated gradients using adjoint method and the true gradient, whether using a ResNet or a HeunNet. Each point corresponds to one parameter.

### 5.1 Validation of our model with step size $\frac{1}{N}$

The ResNet model (1) is different from the classical ResNet because of the $\frac{1}{N}$ term. This makes the model depth aware, and we want to study the impact of this modification on the accuracy on CIFAR and ImageNet. We first train a ResNet-101 [20] on CIFAR-10 and ImageNet using the same hyper-parameters. Experimental details are in appendix B and results are summarized in table 1, showing that the explicit addition of the step size $\frac{1}{N}$ does not affect accuracy. In strike contrast, the

Table 1: Test accuracy (ResNet-101)

|  | ResNet-101 | Ours |
| --- | --- | --- |
| CIFAR-10 | $95.5 \pm 0.1\%$ | $95.5 \pm 0.1\%$ |
| ImageNet | $77.8\%$ | $77.9\%$ |

classical ResNet rule without the scaling $\frac{1}{N}$ makes the network behave badly at large depth, while it still works well with our scaling $\frac{1}{N}$, as shown in Figure 3 (a). On ImageNet, the scaling $\frac{1}{N}$ also leads to similar test accuracy in the weight tied setting: $72.5\%$ with 4 blocks per layer, $73.2\%$ with 8 blocks per layer and $72\%$ with 16 blocks per layer (mean over 2 runs).

### 5.2 Adjoint method

**New training strategy.** Our results in Prop. 3 and 4 assume uniform bounds in $N$ on our residual functions and their derivatives. We also formally proved in the linear setting that these assumptions hold during the whole learning process if the initial loss is small. A natural idea to start from a small loss is to consider a pretrained model. In addition, we also want our pretrained model to verify assumption 2 so we consider the following setup. On CIFAR (resp. ImageNet) we train a ResNet with 4 (resp. 8) blocks in each layer, where weights are tied within each layer. A first observation is that one can transfer these weights to deeper ResNets without significantly affecting the test accuracy of

Table 2: Test accuracy (ResNet)

|  | Before F.T. | After F.T |
| --- | --- | --- |
| CIFAR-10 | $95.25 \pm 0.2\%$ | $95.65 \pm 0.1\%$ |
| ImageNet | $73.1\%$ | $75.1\%$ |

the model: it remains above $94.5\%$ on CIFAR-10 and $72\%$ on ImageNet. We then *untie the weights* of our models and refine them. More precisely, for CIFAR, we then transfer the weights of our model to a ResNet with 4, 4, 64 and 4 blocks within each layer and fine-tune it only by refining the third layer, using our adjoint method. We display in table 2 the median of the new test accuracy, over 5 runs for the initial pretraining of the model. For ImageNet, we transfer the weights to a ResNet with 100 blocks per layer and fine-tune the whole model with our adjoint method for the residual layers. Results are summarized in table 2. To the best of our knowledge, this is the first time a Neural-ODE like ResNet achieves a test-accuracy of $75.1\%$ on ImageNet.

**Failure in usual settings.** In Prop. 3 we showed under assumption 2, that is if the residuals are bounded and Lipschitz continuous with constant independent of the depth $N$, then the error for computing the activations backward would scale in $\frac{1}{N}$ as well as the error for the gradients (Prop. 4). First, this results shows that the architecture needs to be deep enough, because it scales in $\frac{1}{N}$: for instance, we fail to train a ResNet-101 [20] on the ImageNet dataset using the adjoint method on its third layer (depth 23), as shown in Figure 3 (b).

**Success at large depth.** To further investigate the applicability of the adjoint method for training deeper ResNets, we train a simple ResNet model on the CIFAR data set. First, the input is processed by a $5 \times 5$ convolution with 16 out channels, and the image is down-sampled to a size $10 \times 10$. We then apply a batch norm, a ReLU and iterate relation (1) where $f$ is a pre-activation basic block [21]. We consider the zero residual initialisation: the last batch norm of each basic block is initialized to zero. We consider different values for the depth $N$ and notice that in this setup, the deeper our model is, the better it performs in term of test accuracy. We then compare the performance of our model using a ResNet (forward rule (1)) or a HeunNet (forward rule (8)). We train our networks using either the classical backpropagation or our corresponding proxys using the adjoint method (formulas (6) and (9)). We display the final test accuracy (median over 5 runs) for different values of the depth $N$ in Figure 4. The true backpropagation gives the same curves for the ResNet and the HeunNet. Approximated gradients, however, lead to a large test error at small depth, but give the same performance at large depth, hence confirming our results in Prop. 4 and 6. In addition, at fixed depth, the accuracy when training a HeunNet with the adjoint method is better (or similar at depths 2, 32 and 64) than for the ResNet with the adjoint method. This is to be linked with the two different bounds in Prop. 4 and 6:

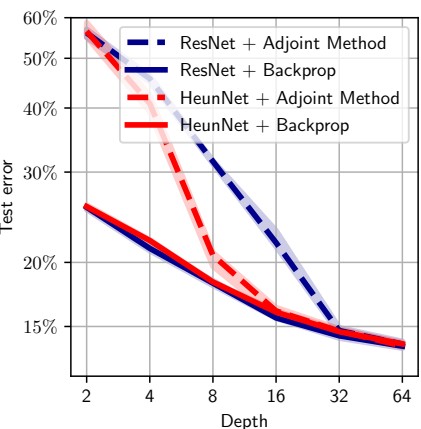

Figure 4: **Comparison** of the best test errors as a function of depth when using Euler or Heun's discretization method with or without the adjoint method.

for the HeunNet, smoothness with depth, which is expected at large depth, according to the theoretical results for the linear case (Prop. 2), implies a faster convergence to the true gradients for the HeunNet than for the ResNet. We finally validate this convergence in Figure 3 (c): the deeper the architecture, the better the approximation on the gradients. In addition, the HeunNet approximates the true gradient better than the ResNet.

## Conclusion, limitations and future works

We propose a methodology to analyze how well a ResNet discretizes a Neural ODE. The positive results predicted by our theory in the linear case are also observed in practice with real architectures: one can successfully use the adjoint method to train ResNets (or even more effectively HeunNets) using very deep architectures on CIFAR, or fine-tune them on ImageNet, without memory cost in the residual layers. However, we also show that for large scale problems such as ImageNet classification from scratch, the adjoint method fails at usual depths.

Our work provides a theoretical guarantee for the convergence to a Neural ODE in the linear setting under a small loss initialization. A natural extension would be to study the non-linear case. In addition, the adjoint method is time consuming, and an improvement would be to propose a cheaper method than a reverse mode traversal of the architecture for approximating the activations.

## Acknowledgments

This work was granted access to the HPC resources of IDRIS under the allocation 2020-[AD011012073] made by GENCI. This work was supported in part by the French government under management of Agence Nationale de la Recherche as part of the "Investissements d'avenir" program, reference ANR19-P3IA-0001 (PRAIRIE 3IA Institute). This work was supported in part by the European Research Council (ERC project NORIA). M. S. thanks Mathieu Blondel and Zaccharie Ramzi for helpful discussions.

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
