# APPENDIX

In Section A we give the proofs of all the propositions, lemmas and the theorem presented in this work.

Section B gives details for the experiments in the paper.

We also give a recap on ResNet architectures in Section C.

## A   Proofs

### A.1   Proof of Prop. 1

Our proof is inspired by [13].

*Proof.* We denote $h = \frac{1}{N}$ and $s_n = nh$. We define

$$\varepsilon_n = x(s_{n+1}) - x(s_n) - h\varphi_\Theta(x(s_n), s_n).$$

We have that $\varphi_\Theta(x(s_n), s_n) = \dot{x}(s_n)$.

Taylor's formula gives

$$x(s_n + h) = x(s_n) + h\dot{x}(s_n) + R_1(h)$$

with $\|R_1(h)\| \leq \frac{1}{2}h^2\|\ddot{x}\|_\infty$. This implies that

$$\|\varepsilon_n\| \leq \frac{1}{2}h^2\|\ddot{x}\|_\infty.$$

The true error we are interested in is the global error $e_n = x(s_n) - x_n$. One has

$$e_{n+1} - e_n = x(s_{n+1}) - x(s_n) + x_n - x_{n+1} = \varepsilon_n + h(\varphi_\Theta(x(s_n), s_n) - \varphi_\Theta(x_n, s_n)).$$

Because $\varphi_\Theta$ is $L$-Lipschitz, this gives $\|e_{n+1} - e_n\| \leq \|\varepsilon_n\| + hL\|e_n\|$ and hence

$$\|e_{n+1}\| \leq (1 + hL)\|e_n\| + \|\varepsilon_n\|.$$

Because $h = \frac{1}{N}$, we have

$$\|e_{n+1}\| \leq (1 + \frac{L}{N})\|e_n\| + \frac{1}{2N^2}\|\ddot{x}\|_\infty.$$

this implies from the discrete Gronwall lemma, since $e_0 = 0$ that

$$\|e_n\| \leq \frac{e^L - 1}{2NL}\|\ddot{x}\|_\infty.$$

Note that we have $\ddot{x} = \partial_s\varphi_\Theta + \partial_x\varphi_\Theta[\varphi_\Theta]$. This gives the desired result. $\qquad\square$

### A.2   Proof of Prop. 2

*Proof.* Recall that we denote $\Pi^N = \prod_{n=1}^N(I_d + \frac{\theta_n^N}{N})$, $\Pi_{:n}^N = (I_d + \frac{\theta_N^N}{N})\ldots(I_d + \frac{\theta_{n+1}^N}{N})$ and $\Pi_{n:}^N = (I_d + \frac{\theta_{n-1}^N}{N})\ldots(I_d + \frac{\theta_1^N}{N})$. We denote $\nabla_n^N = \nabla_{\theta_n^N}L$. One has

$$N\nabla_n^N = \Pi_{:n}^{N\top}(\Pi^N - B)\Sigma\Pi_{n:}^{N\top}.$$

One has as in [51] that

$$\sigma_{max}^2(\Pi_{n:}^N)\sigma_{max}^2(\Pi_{:n}^N)\|\Pi - B\|_\Sigma^2 M \geq N^2\|\nabla_n^N\|^2 \geq \sigma_{min}^2(\Pi_{n:}^N)\sigma_{min}^2(\Pi_{:n}^N)\|\Pi - B\|_\Sigma^2 m.$$

where $\sigma_{max}(A)$ (resp. $\sigma_{min}(A)$) denotes the largest (resp. smallest) singular value of $A$. We first show that $\forall t \in \mathbb{R}_+, \|\theta_n^N(t)\| < \frac{1}{2}$. Denote

$$t^* = \inf\{t \in \mathbb{R}_+, \exists n \in [N-1], \|\theta_n^N(t)\| \geq 1/2\}.$$

One has that $\forall t \in [0, t^*]$, $\sigma^2_{min}(\Pi^N_{:n}) \geq (1 - \frac{1}{2N})^{2(N-n)}$ and $\sigma^2_{min}(\Pi^N_{n:}) \geq (1 - \frac{1}{2N})^{2(n-1)}$ which implies that

$$N^2\|\nabla^N_n(t)\|^2 \geq 2(1 - \frac{1}{2N})^{2N-2}\ell^N(t)m \geq \frac{2}{e}\ell^N(t)m.$$

Similarly one has $N^2\|\nabla^N_n(t)\|^2 \leq 2e\ell^N(t)M$. To summarize, we have the PL conditions for $t \in [0, t^*]$:

$$\frac{2}{e}m\ell^N(t) \leq N^2\|\nabla^N_n(t)\|^2 \leq 2eM\ell^N(t).$$

As a consequence, one has

$$\frac{\mathrm{d}\ell^N}{\mathrm{d}t}(t) = -N\sum_{n=1}^{N}\|\nabla^N_n(t)\|^2 \leq -\frac{2}{e}m\ell^N(t)$$

and thus $\ell^N(t) \leq e^{-\frac{2}{e}mt}\ell^N(0)$.

We have $\theta^N_n(t^*) = \theta^N_n(0) + N\int_0^{t^*}\nabla^N_n$ and $\|\nabla^N_n\| \leq \frac{\sqrt{2eM}}{N}\sqrt{\ell^N}$ so that

$$\|\theta^N_n(t^*)\| \leq \|\theta^N_n(0)\| + \sqrt{2eM}\int_0^{t^*}e^{-\frac{1}{e}mt}\sqrt{\ell^N(0)}dt < \frac{1}{4} + \frac{1}{4} < 1/2.$$

This is absurd by definition of $t^*$ and thus shows that $\forall t \in \mathbb{R}_+$, $\|\theta^N_n(t)\| < \frac{1}{2}$. We also see that $\nabla^N_n$ is integrable so that $\theta^N_n(t)$ admits a limit as $t \to \infty$.

We now show our main result. Note that we have the relationship $(I + \frac{\theta^N_{n+1}}{N})^\top\nabla_{n+1} = \nabla_n(I + \frac{\theta^N_n}{N})^\top$ so that

$$\nabla^N_{n+1} - \nabla^N_n = (I + \frac{\theta^{N\top}_{n+1}}{N})^{-1}(\frac{\nabla^N_n\theta^{N\top}_n - \theta^{N\top}_{n+1}\nabla^N_n}{N}).$$

Because $\|(I + A)^{-1}\| \leq 2$ if $\|A\| \leq \frac{1}{2}$ this gives $\|\nabla^N_{n+1} - \nabla^N_n\| \leq \frac{2}{N}\|\nabla^N_n\|$. Integrating we get

$$\|\theta^N_{n+1}(t) - \theta^N_n(t)\| \leq \|\theta^N_{n+1}(0) - \theta^N_n(0)\| + 2\int_0^t\|\nabla^N_n\|.$$

This gives

$$\|\theta^N_{n+1}(t) - \theta^N_n(t)\| \leq O(\frac{1}{N}) + \frac{1}{N}2\int_0^t\sqrt{2eM\ell^N(0)}e^{-\frac{1}{e}mt}dt = O(\frac{1}{N}),$$

which is the desired result.

$\square$

### A.3 Proof of lemma 1

*Proof.* We adapt a variant of the Ascoli–Arzelà theorem [7]. We showed in Prop. 2 that there exists $C > 0$ that only depends on the initialization such that, $\forall t \geq 0, \forall i \in [N-1]$,

$$\|\theta^N_{n+1}(t) - \theta^N_n(t)\| \leq \frac{C}{N}.$$

This implies that

$$\|\theta^N_j(t) - \theta^N_i(t)\| \leq C\frac{|j-i|}{N}.$$

We also have that

$$\|\theta^N_n(t_1) - \theta^N_n(t_2)\| = \|N\int_{t_1}^{t_2}\nabla^N_n\| \leq C'|t_1 - t_2|$$

with $C' \geq 0$.

Its follows that $\|\psi_N(s_1, t_1) - \psi_N(s_2, t_2)\| \leq \|\psi_N(s_1, t_1) - \psi_N(s_1, t_2)\| + \|\psi_N(s_1, t_2) - \psi_N(s_2, t_2)\|$ and thus

$$(i) \quad \|\psi_N(s_1, t_1) - \psi_N(s_2, t_2)\| \leq C'|t_1 - t_2| + C|s_1 - s_2| + \frac{C}{N}.$$

We also have

$$(ii) \quad \forall N \in \mathbb{N}, \quad \|\psi_N\|_\infty \le \frac{1}{2}.$$

These two properties are essential to prove our lemma. We proceed as follows.

1) First, we denote $((s_j, t_j))_{j \in \mathbb{N}} = (\mathbb{Q} \cap [0,1]) \times \mathbb{Q}_+$. Since we have the uniform bound $(ii)$, we extract using a diagonal extraction procedure a subsequence $\psi_{\sigma(N)}$ such that $\forall j \in \mathbb{N}$,

$$\psi_{\sigma(N)}(s_j, t_j) \to \psi(s_j, t_j)$$

(we denote the limit $\psi(s_j, t_j)$).

2) We show the convergence $\forall s \in [0,1]$ and $t \in \mathbb{R}_+$.

Let $\varepsilon > 0$, $s \in [0,1]$ and $t \in \mathbb{R}_+$. Since $((s_j, t_j))_{j \in \mathbb{N}}$ is dense in $[0,1] \times \mathbb{R}_+$, there exists $k \in \mathbb{N}$ such that $|s_k - s| < \varepsilon$ and $|t_k - t| < \varepsilon$. Let $N, M \in \mathbb{N}$.

We have

$$\|\psi_{\sigma(N)}(s,t) - \psi_{\sigma(M)}(s,t)\| \le \|\psi_{\sigma(N)}(s,t) - \psi_{\sigma(N)}(s_k, t_k)\| + \|\psi_{\sigma(N)}(s_k, t_k) - \psi_{\sigma(M)}(s_k, t_k)\| + \|\psi_{\sigma(M)}(s_k, t_k) - \psi_{\sigma(M)}(s,t)\|$$

so that

$$\|\psi_{\sigma(N)}(s,t) - \psi_{\sigma(M)}(s,t)\| \le 2C\varepsilon + 2C'\varepsilon + \frac{C}{\sigma(N)} + \frac{C}{\sigma(M)} + \|\psi_{\sigma(N)}(s_k, t_k) - \psi_{\sigma(M)}(s_k, t_k)\|.$$

Since $(\psi_{\sigma(N)}(s_k, t_k))_{N \in \mathbb{N}}$ is a Cauchy sequence, this gives for $N, M$ big enough that

$$\|\psi_{\sigma(N)}(s) - \psi_{\sigma(M)}(s,t)\| \le (2(C + C') + 1)\varepsilon$$

and thus $(\psi_{\sigma(N)}(s,t))$ is a Cauchy sequence in $\mathbb{R}^{d \times d}$. As such, it converges and one has

$$\psi_{\sigma(N)}(s,t) \to \psi(s,t).$$

3) Recall that one has

$$\|\psi_{\sigma(N)}(s_1, t_1) - \psi_{\sigma(N)}(s_2, t_2)\| \le C|s_1 - s_2| + \frac{C}{\sigma(N)} + C'|t_1 - t_2|$$

so that letting $N \to \infty$ gives

$$\|\psi(s_1, t_1) - \psi(s_2, t_2)\| \le C|s_1 - s_2| + C'|t_1 - t_2|$$

and $\psi$ is Lipschitz continuous.

4) Let us finally show that the convergence is uniform in $(s,t)$. Let $s \in [0,1]$, $\varepsilon > 0$ and $\delta > 0$ such that if $|s - u| < \delta$, $\forall t \in \mathbb{R}+$,

$$\|\psi_N(s,t) - \psi_N(u,t)\| \le \varepsilon + \frac{C}{N}$$

and $\|\psi(s,t) - \psi(u,t)\| \le \varepsilon$. There exists a finite set of $\{s_j\}_{j=1}^k$ such that

$$[0,1] \subset \cup_{j=1}^k ]s_j - \frac{\delta}{2}, s_j + \frac{\delta}{2}[.$$

For our $s$, there exists $j \in \{1, \ldots, k\}$ such that $\|s - s_j\| \le \delta$.

There also exists $t_0 \ge 0$ such that if $t \ge t_0$,

$$\|\psi_{\sigma(N)}(s,t) - \psi_{\sigma(N)}(s, t_0)\| \le \sqrt{2eM} \int_{t_0}^t e^{-\frac{1}{e}mz} \sqrt{\ell^N(0)} dz \le \varepsilon.$$

We have:

$$\|\psi_{\sigma(N)}(s, t_0) - \psi(s, t_0)\| \le \|\psi_{\sigma(N)}(s, t_0) - \psi_{\sigma(N)}(s_j, t_0)\| + \|\psi_{\sigma(N)}(s_j, t_0) - \psi(s_j, t_0)\| + \|\psi(s_j, t_0) - \psi(s, t_0)\|$$

and thus:

$$\|\psi_{\sigma(N)}(s, t_0) - \psi(s, t_0)\| \le 2\varepsilon + \frac{C}{\sigma(N)} + \max_{j \in \{1, \ldots, k\}} \|\psi_{\sigma(N)}(s_j, t_0) - \psi(s_j, t_0)\| \le 4\varepsilon \text{ for } N$$ big enough.

Finally, $\|\psi_{\sigma(N)}(s,t) - \psi(s,t)\| \le \|\psi_{\sigma(N)}(s,t) - \psi_{\sigma(N)}(s, t_0)\| + \|\psi_{\sigma(N)}(s, t_0) - \psi(s, t_0)\| + \|\psi(s, t_0) - \psi(s,t)\| \le 6\varepsilon$

for $N$ big enough, independently of $t$ and $s$. This concludes the proof. $\qquad \square$

### A.4 Proof of lemma 2

*Proof.* We group terms 2 by 2 in the product $\Pi^{2N}$. One has $(I + \frac{\theta_{2n}^{2N}}{2N})(I + \frac{\theta_{2n-1}^{2N}}{2N}) = (I + \frac{\tilde{\theta}_n^N}{N})$ with

$$\tilde{\theta}_n^N = (\frac{\theta_{2n}^{2N} + \theta_{2n-1}^{2N}}{2} + \frac{\theta_{2n}^{2N}\theta_{2n-1}^{2N}}{4N}),$$

So that $\Pi^{2N} = \tilde{\Pi}^N$ where $\tilde{\Pi}^N$ is defined as $\Pi^N$ with $\tilde{\theta}_n^N$. One has by Prop. 2 that

$$\tilde{\theta}_n^N = \theta_{2n}^{2N} + O(\frac{1}{N}).$$

We will show that $\tilde{\theta}_n^N = \theta_n^N + O(\frac{1}{N})$.

Let $D_n^N = \|\theta_n^N - \tilde{\theta}_n^N\|$ and $D^N = \frac{1}{N}\sum_{n=1}^N D_n$. We have

$$2D_n^N \dot{D}_n^N = -N\langle \nabla_n^N - \tilde{\nabla}_n^N, \theta_n^N - \tilde{\theta}_n^N \rangle.$$

In addition, we have

$$N(\nabla_n^N - \tilde{\nabla}_n^N) = \Pi_{:n}^{N\top}(\Pi^N - B)\Sigma\Pi_{n:}^{N\top} - \tilde{\Pi}_{:n}^{N\top}(\tilde{\Pi}^N - B)\Sigma\tilde{\Pi}_{n:}^{N\top}$$

so that

$$N(\nabla_n^N - \tilde{\nabla}_n^N) = (\Pi_{:n}^N - \tilde{\Pi}_{:n}^N)^\top(\Pi^N - B)\Sigma\Pi_{n:}^{N\top} + \tilde{\Pi}_{:n}^{N\top}(\Pi^N - B)\Sigma(\Pi_{n:}^N - \tilde{\Pi}_{n:}^N)^\top + \tilde{\Pi}_{:n}^{N\top}(\Pi^N - \tilde{\Pi}^N)\Sigma\tilde{\Pi}_{n:}^{N\top}.$$

Note also that since the Jacobian of $(\theta_1, .., \theta_N) \to \Pi^N$ is

$$J_{(\theta_1,..,\theta_N)}(H_1, .., H_N) = \frac{1}{N}\sum_{n=1}^N \Pi_{:n}^N H_n \Pi_{n:}^N$$

and the $\theta_n^N$'s are such that $\|\theta_n^N\| \leq \frac{1}{2}$, there exists a constant $K > 0$ such that $\|\Pi_{:n}^N - \tilde{\Pi}_{:n}^N\| \leq KD^N$. Again because $\|\theta_n^N\| \leq \frac{1}{2}$ and $\|\tilde{\theta}_n^N\| \leq \frac{1}{2}$, this gives

$$N\|\nabla_n - \tilde{\nabla}_n\| \leq \alpha KD^N \sqrt{\ell^N} + \beta\|\Pi^N - \tilde{\Pi}^N\|$$

for some constants $\alpha$, $\beta$. Finally, we have

$$\dot{D}_n^N \leq \frac{1}{2}(\alpha KD^N \sqrt{\ell^N} + \beta\|\Pi^N - \tilde{\Pi}^N\|)$$

which gives $\forall t$

$$2D_n^N(t) \leq \alpha K \int_0^t D^N \sqrt{\ell^N} + \beta \int_0^t \|\Pi^N - \tilde{\Pi}^N\| + O(\frac{1}{N}). \tag{10}$$

We now focus on the $\beta$ term involving $\|\Pi^N - \tilde{\Pi}^N\|$. Denote $\Delta^N = \Pi^N - \tilde{\Pi}^N$. One has

$$\dot{\Delta}^N = -\frac{1}{N}(\sum_{n=1}^N \Pi_{:n}^N \Pi_{:n}^{N\top}(\Pi^N - B)\Sigma\Pi_{n:}^{N\top}\Pi_{n:}^N + \tilde{\Pi}_{:n}^N \tilde{\Pi}_{:n}^{N\top}(\tilde{\Pi}^N - B)\Sigma\tilde{\Pi}_{n:}^{N\top}\tilde{\Pi}_{n:}^N),$$

and equivalently:

$$\dot{\Delta}^N = -\frac{1}{N}(\sum_{n=1}^N [\Pi_{:n}^N \Pi_{:n}^{N\top} - \tilde{\Pi}_{:n}^N \tilde{\Pi}_{:n}^{N\top}](\Pi^N - B)\Sigma\Pi_{n:}^{N\top}\Pi_{n:}^N + \tilde{\Pi}_{:n}^N \tilde{\Pi}_{:n}^{N\top}(\Pi^N - B)\Sigma[\Pi_{n:}^{N\top}\Pi_{n:}^N - \tilde{\Pi}_{n:}^{N\top}\tilde{\Pi}_{n:}^N] + \tilde{\Pi}_{:n}^N \tilde{\Pi}_{:n}^{N\top}(\Pi^N - \tilde{\Pi}^N)\Sigma\tilde{\Pi}_{n:}^{N\top}\tilde{\Pi}_{n:}^N).$$

Note that similarly to $\|\Pi^N - \tilde{\Pi}^N\|$ there exist $K'$ such that $\|\Pi_{:n}^N \Pi_{:n}^{N\top} - \tilde{\Pi}_{:n}^N \tilde{\Pi}_{:n}^{N\top}\| \leq K'D$ so that

$$\|\dot{\Delta}^N + \frac{1}{N}\sum_{n=1}^N \tilde{\Pi}_{:n}^N \tilde{\Pi}_{:n}^{N\top}\Delta^N \tilde{\Pi}_{n:}^{N\top}\tilde{\Pi}_{n:}^N\| \leq aK'D\sqrt{\ell^N}.$$

Let us denote by $H$ the operator:

$$H(\Delta) = \frac{1}{N}\sum_{n=1}^N \tilde{\Pi}_{:n}^N \tilde{\Pi}_{:n}^{N\top}\Delta\Sigma\tilde{\Pi}_{n:}^{N\top}\tilde{\Pi}_{n:}^N.$$

Our (PL) conditions precisely write $-\Delta^\top H(\Delta) \le -\lambda \|\Delta\|^2$ for some $\lambda > 0$. Let $\varphi^N = \frac{1}{2}\|\Delta^N\|^2$. One has

$$\frac{\mathrm{d}\varphi^N}{\mathrm{d}t} = \langle \Delta^N, \dot{\Delta}^N + H(\Delta^N) \rangle - \langle \Delta^N, H(\Delta^N) \rangle$$

so that

$$\frac{\mathrm{d}\varphi^N}{\mathrm{d}t} \le (aK'D\sqrt{\ell^N})\sqrt{2\varphi^N} - 2\lambda\varphi^N.$$

Since $\|\Delta^N\| = \sqrt{2\varphi^N}$ we get

$$\frac{\mathrm{d}\|\Delta^N\|}{\mathrm{d}t} = \frac{\frac{\mathrm{d}\varphi^N}{\mathrm{d}t}}{\sqrt{2\varphi^N}}.$$

We finally have

$$\frac{\mathrm{d}\|\Delta^N\|}{\mathrm{d}t} \le aK'D^N\sqrt{\ell^N} - \lambda\|\Delta^N\|.$$

Integrating, we get

$$\|\Delta^N(t)\| \le -\lambda \int_0^t \|\Delta^N\| + \int_0^t aK'D^N\sqrt{l^N} + O(\frac{1}{N})$$

and then

$$\int_0^t \|\Delta^N\| \le \frac{1}{\lambda}\int_0^t aK'D^N\sqrt{l^N} + O(\frac{1}{N}).$$

Plugging this into (10) leads to

$$0 \le 2D_n^N(t) \le \alpha K \int_0^t D^N\sqrt{\ell^N} + \frac{\beta}{\lambda}\int_0^t aK'D^N\sqrt{\ell^N} + O(\frac{1}{N}).$$

Let $n(t)$ be such that $D_{n(t)}^N(t) = \max_{i \in [1,N]} D_i^N(t)$. We have

$$0 \le 2D_{n(t)}^N(t) \le \mu \int_0^t D_{n(\tau)}^N(\tau)\sqrt{\ell^N(\tau)}d\tau + O(\frac{1}{N})$$

for some constant $\mu > 0$. And since $\sqrt{\ell^N}$ is integrable, we get by Gronwall's inequality that $D_n^N = O(\frac{1}{N}) \; \forall n \in [1,N]$. We showed:

$$\theta_{2n}^{2N} = \theta_n^N + O(\frac{1}{N}).$$

$\square$

## A.5 Proof of Th. 1

We first prove the following lemma 3 before proving Th. 1.

**Lemma 3.** *Under the assumptions of Th. 1, let $\sigma$ be such that $\psi_{\sigma(N)} \to \psi_\sigma$ uniformly (in $\|.\|_\infty$ w.r.t $(s,t)$). Then one has $\Pi^{\sigma(N)}(t) \to \Pi(t)$ uniformly (in $t$) where $\Pi(t)$ maps $x_0$ to the solution at time $1$ of the Neural ODE $\frac{\mathrm{d}x}{\mathrm{d}s} = \psi_\sigma(s,t)x(s)$ with initial condition $x_0$.*

*Proof.* Consider for $x_0 \in \mathbb{R}^d$ with $\|x_0\| = 1$ the discrete scheme

$$x_{n+1} = x_n + \frac{1}{\sigma(N)}\theta_n^{\sigma(N)}(t)x_n,$$

the ODE

$$\frac{\mathrm{d}x}{\mathrm{d}s} = \psi_\sigma(s,t)x(s),$$

and the Euler scheme with time step $\frac{1}{\sigma(N)}$ for its discretization

$$y_{n+1} = y_n + \frac{1}{\sigma(N)}\psi_\sigma(\frac{n}{\sigma(N)}, t)y_n.$$

We know by Prop. 1, since $x_0$ has unit norm that

$$\|x(\frac{n}{\sigma(N)}) - y_n\| \leq \frac{e^{\frac{1}{2}} - 1}{\sigma(N)} \|\partial_s \psi_\sigma(.,t) + \psi_\sigma^2(.,t)\|_\infty^{K \times [0,1]}$$

where $K$ is a compact that contains all the trajectory starting from any unit norm initial condition. Since $\forall t \in \mathbb{R}_+$, $\|\partial_s \psi_\sigma(s,t)\| \leq C$ and $\|\psi_\sigma(s,t)^2\| \leq \frac{1}{2}$, there exists $\tilde{C} > 0$ and independent of $t$ such that

$$\|x(\frac{n}{\sigma(N)}) - y_n\| \leq \frac{\tilde{C}}{\sigma(N)}$$

Now, let $e_n = y_n - x_n$. We have

$$e_{n+1} = e_n(1 + \frac{1}{\sigma(N)}\psi_\sigma(\frac{n}{\sigma(N)},t)) + \frac{1}{\sigma(N)}(\psi_\sigma(\frac{n}{\sigma(N)},t) - \psi_{\sigma(N)}(\frac{n}{\sigma(N)},t))x_n.$$

Since $\|\theta_n^N\| \leq \frac{1}{2}$ and $x_0$ has unit norm, there exists $M > 0$ independent of $x_0$ such that, $\forall n$ and $N$, $\|x_n\| \leq M$. Thus

$$\|e_{n+1}\| \leq \|e_n\|(1 + \frac{1}{2\sigma(N)}) + \frac{1}{\sigma(N)} \sup_{(s,t) \in [0,1] \times \mathbb{R}_+} \|\psi_\sigma(s,t) - \psi_{\sigma(N)}(s,t)\|M.$$

The fact that $\sup_{(s,t) \in [0,1] \times \mathbb{R}_+} \|\psi_\sigma(s,t) - \psi_{\sigma(N)}(s,t)\| \to 0$ (uniform convergence of $\psi_{\sigma(N)}$ to $\psi_\sigma$) along with the discrete Gronwall's lemma leads to $\|e_n\| = o(1)$ independent of $t$ and $x_0$. More precisely,

$$\sup_{t \in \mathbb{R}_+, x_0 \in \mathbb{R}^d, \|x_0\|=1} \|\Pi^{\sigma(N)}(t)x_0 - \Pi(t)x_0\| \to 0$$

as $N \to \infty$. We obtain the uniform convergence with $t$.

$\square$

We can now prove our Th. 1.

*Proof.* Consider $(\psi_{\sigma(N)})_N$ a sub-sequence of $(\psi_N)_N$ as in lemma 1 that converges to some $\psi_\sigma$.

1) We first prove the uniqueness of the limit.

We want to show that $\psi_\sigma$ does not depend on $\sigma$. This will imply the uniqueness of any accumulation point of the relatively compact sequence $(\psi_N)_N$ and thus its convergence.

We have $\forall s \in [0,1]$,

$$\partial_t \psi_{\sigma(N)}(s,t) = -\Pi_{:\lfloor \sigma(N)s \rfloor}^{\sigma(N)\top}(t)(\Pi^{\sigma(N)}(t) - B)\Pi_{\lfloor \sigma(N)s \rfloor:}^{\sigma(N)\top}(t).$$

As $N \to \infty$, we have thanks to lemma 3 that the right hand term converges uniformly to

$$-\Pi_{:s}^\top(t)(\Pi(t) - B)\Pi_{s:}^\top(t)$$

where $\Pi$ maps $x_0$ to the solution at time 1 of the Neural ODE $\frac{dx}{ds} = \psi_\sigma(s,t)x(s)$ with initial condition $x_0$, $\Pi_{:s}(t)$ maps $x_0$ to the solution at time $s$ of the Neural ODE $\frac{dx}{ds} = \psi_\sigma(s,t)x(s)$ with initial condition $x_0$ and $\Pi_{s:}(t)$ maps $x_0$ to the solution at time $1-s$ of the Neural ODE $\frac{dx}{ds} = \psi_\sigma(s,t)x(s)$ with initial condition $x_0$.

This uniform convergence makes it possible to consider the limit ODE as $N \to \infty$:

$$\partial_t \psi_\sigma(.,t) = F(\psi_\sigma(.,t)), \quad \psi_\sigma(.,0) = 0_{d \times d} \tag{11}$$

where $\forall s \in [0,1]$,

$$F(\psi_\sigma(s,t)) = -\Pi_{:s}^\top(t)(\Pi(t) - B)\Pi_{s:}^\top(t).$$

We now show that $F$ is Lipschitz continuous which will guarantee uniqueness through the Picard–Lindelöf theorem. Recall that we have $\forall (s,t) \in [0,1] \times \mathbb{R}_+$:

$$\|\psi_\sigma(s,t)\| \leq \frac{1}{2}.$$

Let $\psi_1, \psi_2$ with $\|\psi_1(s,t)\| \leq \frac{1}{2}$ and $\|\psi_2(s,t)\| \leq \frac{1}{2}$ and $\Pi_1(t), \Pi_2(t)$ the corresponding flows.

Let $x_0$ in $\mathbb{R}^d$ with unit norm, $x_1$ (resp. $x_2$) be the solutions of $\frac{\mathrm{d}x}{\mathrm{d}s} = \psi_1(s,t)x(s)$ (resp. $\frac{\mathrm{d}x}{\mathrm{d}s} = \psi_2(s,t)x(s)$) with initial condition $x_0$. Let $y = x_1 - x_2$.

One has $\Pi_1(t)x_0 = x_1(1)$ and $\Pi_2(t)x_0 = x_2(1)$. One has $\dot{y} = \psi_1 x_1 - \psi_2 x_2 = \psi_2 y + (\psi_1 - \psi_2)x_1$. Hence, since $y(0) = 0$, $\|y(s)\| \leq \int_0^s \|\psi_2\|\|y\| + \|\psi_1 - \psi_2\|_\infty \|x_1\|_\infty$, we have

$$\|y(s)\| \leq \frac{1}{2}\|y(s)\| + \|\psi_1 - \psi_2\|_\infty . \|\Pi_1(t)\|$$

and since $\forall t \in \mathbb{R}_+$, $\|\Pi_1(t)\| \leq 2e$ we get

$$\|\Pi_1(t)x_0 - \Pi_2(t)x_0\| = \|y(1)\| \leq \alpha\|\psi_1 - \psi_2\|_\infty$$

for some $\alpha > 0$. The same arguments go for $\Pi_{:s}$ and $\Pi_{s:}$.

Since we only consider maps $\psi_\sigma$ such that $\|\psi_\sigma(s,t)\| \leq \frac{1}{2}$, this implies that the product is also Lipschitz and thus $F$ is Lipschitz. This guarantees the uniqueness of a solution $\psi$ to the Cauchy problem and we have that $\psi_N \to \psi$ uniformly.

2) We now turn to the convergence speed.

We have $\|\psi_{2N} - \psi_N\| \leq \frac{D}{N}$ for some $D > 0$ thanks to lemma 2. For $k \in \mathbb{N}$, we have that

$$\|\psi_{2^k N} - \psi_N\| \leq \sum_{i=0}^{k-1} \|\psi_{2^{i+1}N} - \psi_{2^i N}\| \leq \frac{D}{N}\sum_{i=0}^{k-1}\frac{1}{2^i} \leq \frac{2D}{N}.$$

Letting $k \to \infty$ finally gives $\|\psi - \psi_N\| \leq \frac{2D}{N}$.

$\square$

## A.6 Proof of Prop. 3

*Proof.* We denote $r_n = \tilde{x}_n - x_n$.

One has $r_N = 0$ and

$$r_n = \tilde{x}_{n+1} - \frac{1}{N}f(\tilde{x}_{n+1}, \theta_n^N) - x_{n+1} + \frac{1}{N}f(x_n, \theta_n^N),$$

that is

$$r_n = r_{n+1} + \frac{1}{N}(f(x_{n+1} - \frac{1}{N}f(x_n, \theta_n^N), \theta_n^N) - f(\tilde{x}_{n+1}, \theta_n^N)).$$

Since

$$f(x_{n+1} - \frac{1}{N}f(x_n, \theta_n^N), \theta_n^N) = f(x_{n+1}, \theta_n^N) - \frac{1}{N}\partial_x f(x_{n+1}, \theta_n^N)[f(x_n, \theta_n^N)] + O(\frac{1}{N^2})$$

this gives

$$r_n = r_{n+1} + \frac{1}{N}(f(x_{n+1}, \theta_n^N) - f(\tilde{x}_{n+1}, \theta_n^N)) - \frac{1}{N^2}\partial_x f(x_{n+1}, \theta_n^N)[f(x_n, \theta_n^N)] + O(\frac{1}{N^3}).$$

Denoting

$$K_N = \sup_{n \in [N-1]} \|\partial_x f(., \theta_n^N)\|_\infty^K \|f(., \theta_n^N)\|_\infty^K,$$

we have the following inequality:

$$\|r_n\| \leq (1 + \frac{L_f}{N})\|r_{n+1}\| + \frac{1}{N^2}K_N + O(\frac{1}{N^3})$$

and since $r_N = 0$, the discrete Gronwall lemma leads to $\|r_n\| \leq \frac{e^{L_f}-1}{L_f N}K_N + O(\frac{1}{N^2})$. In addition, one has $K_N \leq L_f C_f$ so that

$$\|r_n\| \leq \frac{e^{L_f}-1}{N}C_f + O(\frac{1}{N^2}).$$

$\square$

## A.7 Proof of Prop. 4

*Proof.* 1) We first control the error made in the gradient with respect to activations.

Denote

$$g_n = \nabla_{\tilde{x}_n} L - \nabla_{x_n} L.$$

One has using formulas (3) and (7) that

$$g_n = g_{n+1} + \frac{1}{N}(\partial_x f(\tilde{x}_n, \theta_n^N) - \partial_x f(x_n, \theta_n^N))^\top \nabla_{\tilde{x}_{n+1}} L + \frac{1}{N}\partial_x f(x_n, \theta_n^N)^\top g_{n+1}.$$

Since

$$\|\partial_x f(x_n, \theta_n^N)^\top g_{n+1}\| \le L_f \|g_{n+1}\|$$

and because

$$\|(\partial_x f(\tilde{x}_n, \theta_n^N) - \partial_x f(x_n, \theta_n^N))^\top \nabla_{\tilde{x}_{n+1}} L\| \le L_{df} \|\tilde{x}_n - x_n\| g,$$

where $g$ is a bound on $\nabla_{\tilde{x}_{n+1}} L$, we conclude by using Prop. 3 and the discrete Gronwall's lemma.

2) We can now control the gradients with respect to the parameters $\theta_n^N$'s.

Denote

$$t_n = \tilde{\nabla}_{\theta_n^N} L - \nabla_{\theta_n^N} L.$$

We have

$$N t_n = -[\partial_\theta f(x_n, \theta_n^N) - \partial_\theta f(\tilde{x}_n, \theta_n^N)]^\top \nabla_{x_n} L - [\partial_\theta f(\tilde{x}_n, \theta_n^N)]^\top g_n.$$

Hence $N\|t_n\| \le L_\theta \|x_n - \tilde{x}_n\| g + C_\theta \|g_n\|$ where $g$ is a bound on $\nabla_{x_n} L$.

Using our bound on $\|g_n\|$ and Prop. 3 we get

$$N\|t_n\| \le \frac{L_\theta(e^{L_f} - 1)g C_f}{N} + \frac{(e^{L_f} - 1)(e^{L_f} - 1)C_f L_{df} g C_\theta}{L_f N} + O(\frac{1}{N^2})$$

and thus

$$t_n = O(\frac{1}{N^2}).$$

$\square$

## A.8 Proof of Prop. 5

In the following, we let for short $f_n(x) = f(x, \theta_n^N)$, and we define

$$\varphi_n(x) = \frac{1}{2}\left(f_n(x) + f_{n+1}(x + \frac{1}{N}f_n(x))\right) \text{ and } \psi_n(x) = \frac{1}{2}\left(f_{n+1}(x) + f_n(x - \frac{1}{N}f_{n+1}(x))\right) \tag{12}$$

so that Heun's forward and backward equations are

$$x_{n+1} = x_n + \frac{1}{N}\varphi_n(x_n) \text{ and } \tilde{x}_n = \tilde{x}_{n+1} - \frac{1}{N}\psi_n(\tilde{x}_{n+1}).$$

We have the following lemma that quantifies the reconstruction error over one iteration:

**Lemma 4.** *For $x \in \mathbb{R}$, we have as $N$ goes to infinity*

$$\psi_n(x + \frac{1}{N}\varphi_n(x)) - \varphi_n(x) = \frac{1}{4N}(J_{n+1}(x) - J_n(x))[f_{n+1}(x) - f_n(x)] + O(\frac{1}{N^2}),$$

*where $J_n = \partial_x f_n(x)$ is the Jacobian of $f_n$.*

*Proof.* As $N$ goes to infinity, we have the following expansions of (12):

$$\varphi_n(x) = \frac{1}{2}(f_n(x) + f_{n+1}(x)) + \frac{1}{2N}J_{n+1}(x)[f_n(x)] + O(\frac{1}{N^2}),$$

$$\psi_n(x) = \frac{1}{2}(f_n(x) + f_{n+1}(x)) - \frac{1}{2N}J_n(x)[f_{n+1}(x)] + O(\frac{1}{N^2}).$$

As a consequence, we have

$$\psi_n(x + \frac{1}{N}\varphi_n(x)) = \frac{1}{2}(f_n(x) + f_{n+1}(x)) - \frac{1}{2N}J_n(x)[f_{n+1}(x)]$$
$$+ \frac{1}{4N}(J_n(x)[f_n(x) + f_{n+1}(x)] + J_{n+1}(x)[f_n(x) + f_{n+1}(x)]) + O(\frac{1}{N^2}).$$

Putting everything together, we find that the zero-th order in $\psi_n(x + \frac{1}{N}\varphi_n(x)) - \varphi_n(x)$ cancels, and that the first order simplifies to $\frac{1}{4N}(J_{n+1}(x) - J_n(x))[f_{n+1}(x) - f_n(x)]$. □

We now turn the the proof of the main proposition:

*Proof.* We let $r_n = \tilde{x}_n - x_n$ the reconstruction error. We have $r_N = 0$, and we find

$$r_n = \tilde{x}_n - x_n \tag{13}$$

$$= \tilde{x}_{n+1} - \frac{1}{N}\psi_n(\tilde{x}_{n+1}) - x_{n+1} + \frac{1}{N}\varphi_n(x_n) \tag{14}$$

$$= r_{n+1} - \frac{1}{N}(\psi_n(\tilde{x}_{n+1}) - \psi_n(x_{n+1})) - \frac{1}{N}(\psi_n(x_{n+1}) - \varphi_n(x_n)). \tag{15}$$

Using the triangle inequality, and the $L'_f$−Lispchitz continuity of $\psi_n$, we get

$$\|r_n\| \le (1 + \frac{L'_f}{N})\|r_{n+1}\| + \frac{1}{N}\|\psi_n(x_{n+1}) - \varphi_n(x_n)\|.$$

The last term is controlled with the previous Lemma 4:

$$\|\psi_n(x_{n+1}) - \varphi_n(x_n)\| \le \frac{1}{4N}\|(J_{n+1}(x_n) - J_n(x_n))[f_{n+1}(x_n) - f_n(x_n)]\| + O(\frac{1}{N^2}) \tag{16}$$

$$\le \frac{C'_f \Delta_\theta^N}{N} + O(\frac{1}{N^2}). \tag{17}$$

We therefore get the recursion

$$\|r_n\| \le (1 + \frac{L'_f}{N})\|r_{n+1}\| + \frac{C'_f \Delta_\theta^N}{N^2} + O(\frac{1}{N^3}).$$

Unrolling the recursion gives,

$$\|r_n\| \le \frac{(e^{L'_f} - 1)C'_f}{L'_f N}\Delta_\theta^N + O(\frac{1}{N^2}).$$

□

## A.9   Proof of Prop. 6

*Proof.* 1) We first control the error made in the gradient with respect to activations. We have the following recursions:

$$\nabla_{x_n}L = (I + \frac{1}{N}\partial_x\varphi_n(x_{n+1}))^\top \nabla_{x_{n+1}}L \text{ and } \nabla_{\tilde{x}_n}L = (I + \frac{1}{N}\partial_x\varphi_n(\tilde{x}_{n+1}))^\top \nabla_{\tilde{x}_{n+1}}L$$

Letting $r'_n = \nabla_{x_n}L - \nabla_{\tilde{x}_n}L$, we have

$$r'_n = r'_{n+1} + \frac{1}{N}\partial_x\varphi_n(x_{n+1})^\top r'_{n+1} + \frac{1}{N}(\partial_x\varphi_n(x_{n+1}) - \partial_x\varphi_n(\tilde{x}_{n+1}))^\top \nabla_{\tilde{x}_{n+1}}L$$

Therefore, using the triangle inequality, and letting $g$ a bound on the norm of the gradients $\nabla_{\tilde{x}_{n+1}}L$ and $\Delta$ a Lipschitz constant of $\partial_x\varphi_n$, we find

$$\|r'_n\| \le (1 + \frac{L'_f}{N})\|r'_{n+1}\| + \frac{1}{N}g\Delta\|x_{n+1} - \tilde{x}_{n+1}\|$$

The last term is controled with the previous proposition, and we find

$$\|r'_n\| \leq (1 + \frac{L'_f}{N})\|r'_{n+1}\| + \frac{(e^{L'_f} - 1)C'_f g\Delta}{L'_f N^2}\Delta_\theta^N + O(\frac{1}{N^3}),$$

which gives by unrolling:

$$\|r'_n\| \leq \frac{(e^{L'_f} - 1)^2 C'_f g\Delta}{L'^2_f N}\Delta_\theta^N + O(\frac{1}{N^2}).$$

2) We can now control the gradients with respect to parameters. Since Heun's method involves parameters $\theta_n^N$ both for the computation of $x_n$ and $x_{n+1}$, the gradient formula is slightly more complicated than for the classical ResNet. It is the sum of two terms, the first one $\nabla^1_{\theta_n^N} L$ corresponding to iteration $n$ and the second one $\nabla^2_{\theta_n^N} L$ corresponding to iteration $n-1$.

We have

$$\nabla^1_{\theta_n^N} L = \frac{1}{2N}\left(\partial_\theta f(x_n, \theta_n^N) + \frac{1}{N}\partial_x f(y_n, \theta_{n+1}^N)\partial_\theta f(x_n, \theta_n^N)\right)^\top \nabla_{x_n} L$$

and

$$\nabla^2_{\theta_n^N} L = \frac{1}{2N}\left(\partial_\theta f(y_{n-1}, \theta_{n-1}^N)\right)^\top (I + \frac{1}{N}\partial_x f(x_{n-1}, \theta_{n-1}^N))^\top \nabla_{x_{n-1}} L.$$

The gradient $\nabla_{\theta_n^N} L$ is finally

$$\nabla_{\theta_n^N} L = \nabla^1_{\theta_n^N} L + \nabla^2_{\theta_n^N} L.$$

Overall, these equations map the activations $x_n$ and $x_{n-1}$, and the gradients $\nabla_{x_{n-1}} L$ and $\nabla_{x_n} L$ to the gradient $\nabla_{\theta_n^N}$, which we rewrite as

$$\nabla_{\theta_n^N} L = \Psi(x_n, x_{n-1}, \nabla_{x_n} L, \nabla_{x_{n-1}} L),$$

where the function $\Psi$ is explicitly defined by the above equations. With the memory-free backward pass, the gradient is rather estimated as

$$\tilde{\nabla}_{\theta_n^N} L = \Psi(\tilde{x}_n, \tilde{x}_{n-1}, \nabla_{\tilde{x}_n} L, \nabla_{\tilde{x}_{n-1}} L).$$

The function $\Psi$ is Lispchitz-continuous since all functions involved in its composition are Lipschitz-continuous and the activations belong to a compact set, and its Lipschitz constant scales as $\frac{1}{N}$. We write its Lipschitz constant as $\frac{L_\Psi}{N}$, and we get:

$$\|\nabla_{\theta_n^N} L - \tilde{\nabla}_{\theta_n^N} L\| = \|\Psi(x_n, x_{n-1}, \nabla_{x_n} L, \nabla_{x_{n-1}} L) - \Psi(\tilde{x}_n, \tilde{x}_{n-1}, \nabla_{\tilde{x}_n} L, \nabla_{\tilde{x}_{n-1}} L)\| \qquad (18)$$

$$\leq \frac{L_\Psi}{N}(\|x_n - \tilde{x}_n\| + \|x_{n-1} - \tilde{x}_{n-1}\| + \|\nabla_{x_n} L - \nabla_{\tilde{x}_n} L\| + \|\nabla_{x_{n-1}} L - \nabla_{\tilde{x}_{n-1}} L\|).$$
$$\qquad (19)$$

Using the previous propositions, we get:

$$\|\nabla_{\theta_n^N} L - \tilde{\nabla}_{\theta_n^N} L\| = O(\frac{\Delta_\theta^N}{N^2} + \frac{1}{N^3}).$$

$\square$

# B    Experimental details

In all our experiments, we use Nvidia Tesla V100 GPUs.

## B.1 CIFAR

For our experiments on CIFAR-10 (training from scratch), we used a batch-size of $128$ and we employed SGD with a momentum of $0.9$. The training was done over $200$ epochs. The initial learning rate was $0.1$ and we used a cosine learning rate scheduler. A constant weight decay was set to $5 \times 10^{-4}$. Standard inputs preprocessing as proposed in Pytorch [33] was performed.

For our finetuning experiment on CIFAR-10, we used a batch-size of $128$ and we employed SGD with a momentum of $0.9$. The training was done over $5$ epochs. The learning rate was kept constant to $10^{-3}$. A constant weight decay was set to $5 \times 10^{-4}$. Standard inputs preprocessing as proposed in Pytorch was also performed.

For our experiment with our simple ResNet model that processes the input by a $5 \times 5$ convolution with 16 out channels, we used a batch-size of $256$ and we employed SGD with a momentum of $0.9$. The training was done over $90$ epochs. The learning rate was set to $10^{-1}$ and was decayed by a factor 10 every 30 epochs. A constant weight decay was set to $5 \times 10^{-4}$. Standard inputs preprocessing as proposed in Pytorch was also performed.

## B.2 ImageNet

For our experiments on ImageNet (training from scratch), we used a batch-size of $256$ and we employed SGD with a momentum of $0.9$. The training was done over $100$ epochs. The initial learning rate was $0.1$ and was decayed by a factor 10 every 30 epochs. A constant weight decay was set to $10^{-4}$. Standard inputs preprocessing as proposed in Pytorch was performed: normalization, random croping of size $224 \times 224$ pixels, random horizontal flip.

For our finetuning experiment on ImageNet, we used a batch-size of $256$ and we employed SGD with a momentum of $0.9$. The training was done over $3$ epochs. The learning rate was kept constant to $5 \times 10^{-4}$. A constant weight decay was set to $10^{-4}$. Standard inputs preprocessing as proposed in Pytorch was performed: normalization, random croping of size $224 \times 224$ pixels, random horizontal flip.

## C  Architecture details

In computer vision, the ResNet as presented in [20] first applies non residual transformations to the input image: a feature extension convolution that goes to 3 channels to 64, a batch norm, a non-linearity (ReLU) and optionally a maxpooling.

It is then made of $4$ layers (each layer is a series of residual blocks) of various depth, all of which perform residual connections. Each of the $4$ layers works at different scales (with an input with a different number of channels): typically 64, 128, 256 and 512 respectively. There are two types of residual blocks: Basic Blocks and Bottlenecks. Both are made of a successions of convolutions $\mathrm{conv}$, batch normalizations $\mathrm{bn}$ [22] and ReLU non-linearity $\sigma$. For example, a Basic Block iterates (in a pre-activation [21] fashion):

$$x \to x + \mathrm{bn}(\mathrm{conv}(\sigma(\mathrm{bn}(\mathrm{conv}(\sigma(x)))))).$$

Finally, there is a classification module: average pooling followed by a fully connected layer.