# OpenReview forum: "Do Residual Neural Networks discretize Neural Ordinary Differential Equations?"
_NeurIPS.cc/2022/Conference — NeurIPS 2022 Accept_

### Official Review · Reviewer_hARH · 2022-07-08

**Rating:** 7
**Confidence:** 3
**Soundness:** 3 good
**Presentation:** 3 good
**Contribution:** 3 good

**Summary:**

- The paper examines the relationship between and NeuralODEs and ResNets with $N$ layers whose residual functions have a multiplicative coefficient of $1/N$, i.e. with layers of the type $x_{n+1} = x_n + \frac{1}{N}f_{\theta_n}(x_n)$.
- The paper shows that if the initialised weights of a _linear_ ResNet vary smoothly from one layer to the next and other assumptions at training time $t=0$ are satisfied, as $N\rightarrow \infty$, the map learned by the ResNet at any training time $t$ corresponds to the map induced by a (training time-dependent) ODE sending $x(0) \mapsto x(1)$, where $x(1)$ is the solution of this ODE at time $1$.
- The paper shows that the smoothness assumption mentioned above is necessary and this convergence is not otherwise guaranteed.
- The paper investigates how these sort of ResNets can be trained with $O(1)$ memory via a discrete adjoint method producing approximate gradients. It further investigates what the error of this approximation is as a function of $N$ and shows how this error can be reduced by using Heun's method.
- Finally, the experimental section shows ResNets with rate $1/N$ work similarly well in practice as regular ResNets and study how the proposed discrete adjoint can be used to fine-tune deep ResNets with (initially) tied weights.

**Questions:**

- Most importantly, everything in the first bullet point of the weaknesses section.
- In line 328, what do you mean by "the classical ResNet forward rule is not compatible with the usual training procedure of CIFAR"?

**Limitations:**

Yes.

**Strengths And Weaknesses:**

# Strenghts

- Despite the well-known connections between ResNets and NeuralODEs, this paper shows that there are still many interesting aspects to uncover about their relationship. Usually, when this equivalence is discussed, it is typically assumed that $\theta_n = \theta$.  This paper investigates this equivalence in the linear regime when the weights $\theta_n$ vary at each layer $n$, which is much closer to how ResNets are actually constructed and used in practice. Therefore, this improves our understanding of a practically-relevant situation and it is likely that this result will also be relevant to practitioners and not just the theoretically inclined.
- The proposed discrete adjoint method comes with a useful theoretical analysis providing theoretical guarantees regarding the error potential users should expect.
- Although not spectacular or surprising, the successful experiments on reproducing the performance of regular ResNets using ResNets with rate $1/N$ are an important result. It shows that this model that is convenient theoretically is also relevant in practice and essentially it does not affect the usual performance we expect from ResNets. This also makes the theory more relevant for practitioners.
- I like that the paper also explores failure modes in the experimental section for when the assumptions of the results do not hold.
- In the era of "large models", the presented fine-tuning experiments are very relevant and could be a useful tool in the arsenal of anyone training deep ResNets.

# Weaknesses

- I believe the main problem of the paper is the (too) many notational errors or omissions for a theoretical paper. While each in itself is minor, together they accumulate and they do tend to show up in important parts of the paper. So I am adding a list below:
    1. In the bound of Proposition 1, if $L=0$, then $\frac{e^L - 1}{L} = \frac{0}{0}$, which is undefined. So I do not understand the calculations done below where $L=0$. Am I missing something here?
    2. What does the notation $[\cdot]$ mean in the definition of $C_N$ (line 142). The $[\cdot]$ notation is also used in other places. And what does the superscript $K \times [0, 1]$ signify?
    3. Please explain the notation $C^k$ for classes of smooth functions. Many people with ML background might not be familiar with it.
    4. Before Proposition 1, it says ``We now consider a smooth interpolation $\phi_\Theta$ for ResNet''. Which interpolation do you actually use? Two were introduced in the paragraph above. Does it matter?
    5. Below Proposition 1 you mention that the bound is tight. But you show this only for the case when the ODE function depends only on time, which makes L superfluous (ignoring the undefined issue). But is the bound actually tight in general? This question is probably not super relevant for the discussion in the implications paragraph below, but I think it's still an important thing to clarify.
    6. In line 188, $\nabla_n^N(t) = \nabla_{\theta_n^N}L$, there is no $t$ on the right hand side. Should that be $\nabla_{\theta_n^N}L(t)$?
    7. What is the difference between $L(t)$ and $l^N(t)$. Both are used and as far as I understand they denote the same thing.
    8. In Assumption 2, what is $D_x$? As far as I can see, $D_x$ is never defined. Should that be $\partial_x$ instead?
    9. In Assumption 2, $C_{df}$ is defined and never used. Instead, an $L_f$ shows up, that was never introduced. Should $L_f$ be $C_{df}$?
    10. Proposition 6, No need to mention $L_{df}$ if it doesn't show up in the final bound.
    11. Please clarify what you mean by "tied weights". I checked many ResNet papers and it doesn't seem to be standard terminology. I take it to mean $\theta_n = \theta$. But this can be a confusing term because the smoothness assumption you often employ also "ties" the weights of different layers together. Constant weights is a better term IMO.
- Figure 2 is not particularly useful and could show something much more interesting and visualise it much better. First of all, a sequential colour scale is much more appropriate for what it is trying to show because this shows a progression of training steps. A categorical colouring scale does not convey that information. Second, plotting just a few weights in the matrices loses a lot of information. The plot could show instead the distance from the limit $\psi$ as a function of the training time or another aggregated metric of interest. Perhaps the distance between consecutive layers and how it evolves over time could be another idea.
- In the first plot of Figure 3, since this is the median over only 3 runs, it's better to just include all 3 runs (6 runs in total) since it's not a lot. The median drops a lot of information and doesn't say much about the variation. Alternatively, better to show the mean and a confidence interval.
- In line 377, you mention that "the HeunNet ... performs better than the ResNet". This claim is not really backed up by data. It's impossible to deduce that from Figure 4 and we would need to see some numerical data. There are two things that are needed to support that claim: scientific significance (if the difference is just 0.00001%, and it looks very small on the plot, then it's not really better) and statistical significance (what is the standard error?). The claim should either be dropped or more data should be supplied.

======= EDIT ===========

Raised my score to accept after discussion.

---

> ### Author Response · Authors · 2022-08-02
> **Response to reviewer hARH**
>
> We would like to thank Reviewer hARH for all their positive compliments, the relevance of their remarks, and the spotted typos in the text. We updated the paper according to the remarks made by the reviewer.
>
> > Remarks 1 to 11
>
> 1) As L goes to 0 this quantity goes to L/L = 1. This is clarified in the updated version of the paper: line 159.
>
> 2) This has been clarified in the final version of the paper. [.] means that we evaluate the Jacobian at that point, and $K \times [0, 1]$ describes the set {$(x,y) | x \in K, y \in [0,1]$}. We added a “notation” paragraph in the paper to define these notions: lines 132 to 134.
>
> 3) This is clarified in the updated version. It refers to the set of functions that are k times differentiable with their kth differential being continuous.
>
> 4) We clarified that any smooth interpolation works: line 152.
>
> 5) This is a very good question that we are currently investigating.
>
> 6) You are right, thank you. We updated it in line 202.
>
> 7)  L takes as input the parameters $(\theta_1(t), …, \theta_N(t))$.  We here use $l^N$ to denote the corresponding function that only depends on t. : $l^N(t) = L (\theta_1(t), …, \theta_N(t))$.
>
> 8) Yes, thank you, it should be $\partial_x$ instead. We updated it in line 279.
>
> 9) Yes, actually one can use one or the other, since any bound on the derivative is also a Lipschitz constant. This has been clarified and we only use $C_{df}$.
>
> 10) You are right. It has been removed.
>
> 11) Yes, we mean that the weights are constant: $\theta_n = \theta$. We hold this terminology from the deep equilibrium models paper [1]. This is clarified in the updated version in line 71.
>
> [1] Bai, S., Kolter, J. Z., & Koltun, V. (2019). Deep equilibrium models. Advances in Neural Information Processing Systems, 32. https://arxiv.org/abs/1909.01377
>
> > In line 328, what do you mean by "the classical ResNet forward rule is not compatible with the usual training procedure of CIFAR"?
>
> We mean that the classical ResNet rule without the $1/N$ makes the network behave badly at large depth, while it still works well with the $1/N$. This has been clarified in lines 333 to 335.
>
> > Figure 2 is not particularly useful and could show something much more interesting and visualize it much better. First of all, a sequential colour scale is much more appropriate for what it is trying to show because this shows a progression of training steps. A categorical colouring scale does not convey that information. Second, plotting just a few weights in the matrices loses a lot of information. The plot could show instead the distance from the limit ψ as a function of the training time or another aggregated metric of interest. Perhaps the distance between consecutive layers and how it evolves over time could be another idea.
>
> We thank the reviewer for their very good idea regarding this figure. We followed your  recommandations. An updated version of the figure is now in the paper.
>
> > In the first plot of Figure 3, since this is the median over only 3 runs, it's better to just include all 3 runs (6 runs in total) since it's not a lot. The median drops a lot of information and doesn't say much about the variation. Alternatively, better to show the mean and a confidence interval.
>
> We updated the figure with the 3 runs.
>
> > In line 377, you mention that "the HeunNet ... performs better than the ResNet". This claim is not really backed up by data. It's impossible to deduce that from Figure 4 and we would need to see some numerical data. There are two things that are needed to support that claim: scientific significance (if the difference is just 0.00001%, and it looks very small on the plot, then it's not really better) and statistical significance (what is the standard error?). The claim should either be dropped or more data should be supplied.
>
> We clarified in lines 383 to 384 what we mean by "the HeunNet ... performs better than the ResNet". We mean that the accuracy when training a HeunNet with the adjoint method is better than for the ResNet with the adjoint method (same depth). This is clearly illustrated in Fig. 4 (dashed lines), where the red curve (HeunNet) is always under the blue curve (ResNet).

---

> > ### Comment · Reviewer_hARH · 2022-08-05
> > **Response**
> >
> > Thank you for your detailed reply! I am satisfied with most of your replies, with a few exceptions:
> >
> > > We mean that the accuracy when training a HeunNet with the adjoint method is better than for the ResNet with the adjoint method (same depth)
> >
> > This is clearly not true from the plot. Yes, there is a depth range where HeunNet with adjoint is clearly better than ResNet with adjoint, but this is not the case for all depths (which is what you are claiming). At depth 2-3 and at depth 32-64 the two dashed lines and their errors overlap significantly and seem to perform rather similarly (at least visually). So I think that claim should be adjusted to "HeunNet performs similarly or better" or something similar.
> >
> > > This quantity is also well defined at $L = 0$ since $\frac{e^L - 1}{L} \rightarrow 1$ as $L \rightarrow 0$.
> >
> > What you are saying is that the function has a limit (i.e. approaches a finite value as $L \rightarrow 0$) but the function in your bound is still not defined at $0$, so your bound is not rigorously correct. You should either rewrite your bound to use a function defined differently on $L=0$ and $L > 0$ or, alternatively, rewrite your bound for $L$ layers as the limit of $\frac{e^l - 1}{l}$ as $l \rightarrow L$.
> >
> > > $K \times [0, 1]$ describes the [cartesian product]
> >
> > This is a misunderstanding. I should have clarified better. My question was what does this product signify in the superscript of a norm (i.e. $|| \cdot ||^{K \times [0, 1]}$) in line 156. Does it simply say it is a norm over this space? I did not come across this notation before and I think it should be explained.

---

> > > ### Author Response · Authors · 2022-08-05
> > > **Second response to Reviewer hARH**
> > >
> > > Thank you for your reply. We are glad we answered most of your questions. Regarding the remaining $3$ replies you are not satisfied about, we agree with your remarks and advice and modified the submission accordingly:
> > >
> > > >  This is clearly not true from the plot. Yes, there is a depth range where HeunNet with adjoint is clearly better than ResNet with adjoint, but this is not the case for all depths (which is what you are claiming). At depth 2-3 and at depth 32-64 the two dashed lines and their errors overlap significantly and seem to perform rather similarly (at least visually). So I think that claim should be adjusted to "HeunNet performs similarly or better" or something similar.
> > >
> > > It's true that at depth 2 and 32-64 both models perform similarly. We clarified it in the paper (line 383).
> > >
> > > > What you are saying is that the function has a limit (i.e. approaches a finite value as ) but the function in your bound is still not defined at 0, so your bound is not rigorously correct.
> > >
> > > We adjusted it in the paper (line 157).
> > >
> > > >This is a misunderstanding. I should have clarified better. My question was what does this product signify in the superscript of a norm (i.e. $\| .\|^{K\times[0,1]}$ ) in line 156. Does it simply say it is a norm over this space? I did not come across this notation before and I think it should be explained.
> > >
> > > Sorry for the misunderstanding. Yes, it means we take the supremum over this space. We clarified it in the notation paragraph (line 134).
> > >
> > > Thank you again for your comments and your review.

---

> > > > ### Comment · Reviewer_hARH · 2022-08-05
> > > > **Response**
> > > >
> > > > Thank you for addressing these last comments as well!
> > > >
> > > > I also want to mention that I have read Reviewer's rbYF concerns regarding the fact that the results are "far from the practice". I agree this is true to a certain extent, but I believe this should not stop us from studying these models in simplified settings. I believe that the paper has the merit of studying the connection to Neural ODEs in a setting that is to some extent *closer* to practice (even if still far in absolute terms) by considering the case when the weights are not constant at all layers. At the same time, the paper made some efforts to show that the theoretical model it works with is relevant in practice since it performs similarly to a classic ResNet (without 1/N scaling). This argues that the theoretical model is also relevant in practice. Finally, the deep linear setting has been the main playground of Deep Learning theory for quite some time and even though it is a huge simplification of the non-linear regime, one would still like to have a good understanding of it.
> > > >
> > > > That being said, I am happy to raise my score to accept as I believe all my concerns have been addressed and the clarity of the paper has significantly improved.

---

### Official Review · Reviewer_9X5i · 2022-07-11

**Rating:** 6
**Confidence:** 3
**Soundness:** 3 good
**Presentation:** 2 fair
**Contribution:** 2 fair

**Summary:**

This paper studies the link between Neural ODEs and ResNets from a theoretical point of view.

The authors show that without extra regularity assumptions on the residual block, a ResNet might not converge to a Neural ODE in the limit when the depth goes to infinity. On the positive side, they then show how this issue can be resolved by assuming additional assumptions on the weight initialiatiation and study the case of linear residual blocks.

An additional contribution is a numerical technique allowing to perform backpropagation through ResNets similarly to the adjoint method for Neural ODEs; this has the advantage of not requiring to store activations, and therefore comes with constant memory footprint. The disadvantage is given by inaccurate gradients. Error bounds for recovering activations and gradients are provided, and even improved by using a modification of the proposed adjoint method analogue to the Heun's method for differential equations.






**Questions:**

How much do the results in section 3.2 depend on the choice of loss function?

What are the main theoretical bottlenecks preventing from extending the proofs on section 3.2 from the linear case to the non-linear case? Are there some non-linear activation functions for which this extension could be easier than others?

**Limitations:**

As mentioned above, the main limitation is that the study is limited to the linear case for the residual block, but the authors clearly mention this in their conclusion.

**Strengths And Weaknesses:**

This paper sheds some light on the theoretical aspects relating Neural ODEs and ResNets, which has recently become a popular topic in the deep learning community. Several contributions have been provided in this direction prior to this paper, in particular highlighting the failure, in some cases, of ResNets to approximate ODEs in the infinite depth limit. Therefore, the level of originality of the paper is not very high, but the contributions build upon existing ones and the literature seems appropriately cited. The proposed technique based on the adjoint method to train ResNets without storing activations is interesting and yields improvements in performance on CIFAR-10 and ImageNet.

The main weakness is that the authors only study the linear residual block case.

---

> ### Author Response · Authors · 2022-08-02
> **Response to reviewer 9X5i**
>
> We would like to thank reviewer 9X5i for their review.
>
> > The main weakness is that the authors only study the linear residual block case.
>
> We acknowledge that this is a limitation, but going beyond the linear case seems currently out of reach. For instance, the result closest to our work [1] uses a non linearity but their activation function is smooth and satisfies $f(x$) ~ $x$ as $x \to 0$, and the authors perform a linearization as the depth $N \to \infty$, which is then equivalent to studying the linear case.
> Furthermore, let us stress that we have other contributions not limited to the linear case. In particular, we have a new method (discrete adjoint method) inspired by our theoretical findings, we derive bounds on the error made for approximating gradients in the general case using this method (prop. 3 to 6), and illustrate its success in practice  (Figure 3c and Figure 4).
>
> [1] Cont, R., Rossier, A., & Xu, R. (2022). Convergence and Implicit Regularization Properties of Gradient Descent for Deep Residual Networks. arXiv preprint arXiv:2204.07261. https://arxiv.org/abs/2204.07261
>
> > How much do the results in section 3.2 depend on the choice of loss function?
>
> While for simplicity we wrote these results for the squared-Euclidean loss, they still hold for arbitrary loss function as long it satisfies a Polyak-Łojasiewicz condition.
>
> > What are the main theoretical bottlenecks preventing extending the proofs on section 3.2 from the linear case to the non-linear case?  Are there some non-linear activation functions for which this extension could be easier than others?
>
> An intermediate step would be to study the case where $f(x, \theta)$ is linear in $\theta$ but not in $x$ such as in [1]. In this work, the residual block has linear parametrization while still being nonlinear. While such a network learns non-linear transformations, the gradient flow still seems to have a simple structure because it can be studied in a RKHS.
>
> [1] Barboni, R., Peyré, G., & Vialard, F. X. (2021). Global convergence of ResNets: From finite to infinite width using linear parameterization. arXiv preprint arXiv:2112.05531.  https://arxiv.org/abs/2112.05531

---

### Official Review · Reviewer_rbYF · 2022-07-20

**Rating:** 5
**Confidence:** 4
**Soundness:** 3 good
**Presentation:** 3 good
**Contribution:** 2 fair

**Summary:**

This paper studies the connection between ResNet and Neural ODE. Specifically, (1). this paper investigates the approximation error between the hidden state trajectory of the Resnet and the solution of the Neural ODE interpolating the hidden state of the Resnet. The approximation error can have a non-zero limit in general, and the authors specify a case (small enough initialization, linear blocks) where the error limit is zero. (2). The paper then views the hidden state trajectory of ResNet as a forward Euler method to solve the corresponding NODE and proposes to use the model obtained by the backward Euler method in the backpropagation, which has a lower memory cost. Such a practice is also transferred to the HeunNet with a theoretically lower approximation error bound. The authors further conduct experiments to verify the theoretical findings.

**Questions:**

1. Why do we need to save the memory cost for the hidden state, which is relatively small? Specifically, the dimension of one hidden state (size m) is way smaller than the dimension of the corresponding layer’s parameter (size m^2).
2. What’s the point to approximate the NODE using Resnet (or study this regularization effect)? I expect the authors can comment on the advantage of NODE over ResNet and why we need to study this regularization effect.


**Limitations:**

Please refer to the "weakness" and “questions”.

**Strengths And Weaknesses:**

Strengths:
1.	This paper is well-written
2.	The theoretical analysis is solid, and the authors honestly report the experiment results.

Weakness:
1.	The settings of the theoretical analysis are far from the practice. This paper uses the forward propagation rule x_{t+1}=x_{t} +1/N f(x_t,\theta_t), where N is the depth of neural networks. However, in the latter analysis (e.g. Proposition 1,2), it is assumed that \theta_t has bounded norm (or f has bounded Lipschitz parameter with respect to theta_t). Therefore, if we absorb the 1/N factor into f (or more directly into \theta_t, as in neural networks, f is usually a piece-wise linear function of \theta_t), which is what we do in practice, the initialization scale of \theta can be rather small and decreases with the depth, while in practice the initialization scale usually depends on the width. Theorem 1 requires the parameters between adjacent layers to be close, which is also unrealistic.
2.	The experiment results are not promising even compared to the current state of the arts. For example, in table 2, the test accuracy on Imagenet for Resnet using the adjoint method is not high even compared to the vanilla Resnet [1]. Also, the performance of the adjoint method in Figure 4 is worse than the traditional backpropagation.

[1]. https://paperswithcode.com/sota/image-classification-on-imagenet?tag_filter=3

---

> ### Author Response · Authors · 2022-08-02
> **Response to reviewer rbYF (1/2)**
>
> We would like to thank reviewer rbYF for their review. We appreciate that the reviewer found our theoretical analysis “solid”.
> We now address the questions of the reviewer. In particular, regarding the main criticism of the reviewer, we stress out that the initialization of parameters in very deep ResNets must scale with depth, and that the memory bottleneck in deep ResNets in practical applications really comes from storing the activations and not the parameters.
>
> > The settings of the theoretical analysis are far from the practice.
> This paper uses the forward propagation rule x_{t+1}=x_{t} +1/N f(x_t,\theta_t), where N is the depth of neural networks.
>
> We agree with the reviewer that for usual resnet of moderate size (typically ResNet152), using a constant scaling (1 instead of 1/N) is acceptable. We found that using a scaling 1/N leads to the same performance (Table 1). However, we show that using a scaling 1/N works far better when the weights are tied between layers (Figure 3 (a)).
>
> > However, in the latter analysis (e.g. Proposition 1,2), it is assumed that \theta_t has bounded norm (or f has bounded Lipschitz parameter with respect to theta_t).
> Therefore, if we absorb the 1/N factor into f (or more directly into \theta_t, as in neural networks, f is usually a piece-wise linear function of \theta_t), which is what we do in practice, the initialization scale of \theta can be rather small and decreases with the depth, while in practice the initialization scale usually depends on the width.
>
> We believe the reviewer has in mind the initialization scale for feedforward networks (e.g. [1]). However, for ResNets, the best initialization scale for the parameters have been found to also depend on depth. For instance, in [2], it is said:
>
> “We establish theoretically and empirically that the best initialization variances for residual networks depend on the depth of the network (contrary to the feedforward case), so that common initialization schemes like Xavier or He cannot be optimal”
>
> Actually, one can show, using a similar variance analysis as in [1]  that a constant step size in ResNet leads to an exploding variance phenomenon.
>
> Also, recent work shows that the weights scale with depth experimentally (in $1/N^\beta$) [3].
>
> In addition, for very deep ResNets, such as for approximating Neural ODEs, using a constant step size is not acceptable because there is no limit as $N \to +\infty$ in the two interesting settings studied in the literature and in our paper:
>
> -Smooth initialization (typically 0 or with small $|\theta(0)_{t+1}-\theta(0)_t|$) as we consider, the scaling needs to be O(1/N) to obtain smooth trajectories.
>
> -Random initialization with $\theta(0)_{t+1}$ and $\theta(0)_t$ independent, as considered in [3, 4], in which case the scaling needs to be $O(1/\sqrt{N})$ to obtain continuous trajectory which are rough paths (typically 1/2 holder continuous).
>
> All these considerations are now clarified in the updated version of the paper: lines 212 to 216.
>
> [1] He, K., Zhang, X., Ren, S., & Sun, J. (2015). Delving deep into rectifiers: Surpassing human-level performance on imagenet classification. In Proceedings of the IEEE international conference on computer vision (pp. 1026-1034) https://arxiv.org/abs/1502.01852
>
> [2] Yang, G., & Schoenholz, S. (2017). Mean field residual networks: On the edge of chaos. Advances in neural information processing systems, 30. https://arxiv.org/abs/1712.08969
>
> [3]  Cohen, A. S., Cont, R., Rossier, A., & Xu, R. (2021, July). Scaling properties of deep residual networks. In International Conference on Machine Learning (pp. 2039-2048). PMLR.  https://arxiv.org/abs/2105.12245
>
> [4] Cont, R., Rossier, A., & Xu, R. (2022). Convergence and Implicit Regularization Properties of Gradient Descent for Deep Residual Networks. arXiv preprint arXiv:2204.07261. https://arxiv.org/abs/2204.07261
>
> > Theorem 1 requires the parameters between adjacent layers to be close, which is also unrealistic.
>
> We believe this type of initialization is now becoming mainstream, and corresponds to the Fixup initialization, a popular initialization method for ResNets [1]: the residuals f are initialized to 0. Initializing to 0 (or more generally with close adjacent layers) is necessary to generate smooth trajectories at initialization and during training.
>
> [1] Zhang, H., Dauphin, Y. N., & Ma, T. (2019). Fixup initialization: Residual learning without normalization. arXiv preprint arXiv:1901.09321  https://arxiv.org/abs/1901.09321

---

> > ### Author Response · Authors · 2022-08-02
> > **Response to reviewer rbYF (2/2)**
> >
> >  > The experiment results are not promising even compared to the current state of the arts. For example, in table 2, the test accuracy on Imagenet for Resnet using the adjoint method is not high even compared to the vanilla Resnet [1].
> >
> > This is true, however this corresponds to our proposed discrete adjoint method: gradients are approximated. It performs much better than any previous model using the adjoint method on ImageNet (e.g. [1] Table 2: we do 75% against 70.2 %).
> >
> > [1] Zhuang, J., Dvornek, N. C., Tatikonda, S., & Duncan, J. S. (2021). MALI: A memory efficient and reverse accurate integrator for Neural ODEs. arXiv preprint arXiv:2102.04668. https://openreview.net/pdf?id=blfSjHeFM_e
> >
> >
> > > Also, the performance of the adjoint method in Figure 4 is worse than the traditional backpropagation.
> >
> > This is no longer true at large depth where the performance of our proposed adjoint method is the same as the traditional backpropagation, which is the goal of this experiment. It validates the theoretical findings of propositions 3 to 6.
> >
> > > Why do we need to save the memory cost for the hidden state, which is relatively small? Specifically, the dimension of one hidden state (size m) is way smaller than the dimension of the corresponding layer’s parameter (size m^2).
> >
> > The memory bottleneck really comes from storing the activations in practice, and designing invertible architectures to avoid this issue has been an active field in recent years (e.g. [1] , [2] , [3]).
> > Indeed, samples are not processed one by one but rather in mini batches of size usually between 64 and 1024. In addition, in computer vision applications, convolutional networks are used so that, if one hidden state is of size m, one does not need to store m^2 parameters for the corresponding linear operator.
> > To illustrate this statement, we conducted an additional experiment. We trained a ResNet152 on ImageNet and used a memory profiler. We found out that the memory needed to store the parameters of the model is 220 MiB, whereas the memory needed to store the activations is about 22 GiB for batches of size 128.
> > We clarified it in the updated version of the paper: lines 109 to 118.
> >
> > [1] Gomez, A. N., Ren, M., Urtasun, R., & Grosse, R. B. (2017). The reversible residual network: Backpropagation without storing activations. Advances in neural information processing systems, 30. https://proceedings.neurips.cc/paper/2017/file/f9be311e65d81a9ad8150a60844bb94c-Paper.pdf
> >
> > [2] Sander, M. E., Ablin, P., Blondel, M., & Peyré, G. (2021, July). Momentum residual neural networks. In International Conference on Machine Learning (pp. 9276-9287). PMLR. https://arxiv.org/abs/2102.07870
> >
> > [3] Jacobsen, J. H., Smeulders, A., & Oyallon, E. (2018). i-revnet: Deep invertible networks. arXiv preprint arXiv:1802.07088. https://arxiv.org/abs/1802.07088
> >
> >
> > > What’s the point of approximating the NODE using Resnet (or study this regularization effect)? I expect the authors can comment on the advantage of NODE over ResNet and why we need to study this regularization effect.
> >
> > It is above all a theoretical question to understand the behavior of ResNets in the very deep regime. By approximating a ResNet with a NODE, one can study the properties of these architectures using the arsenal of ODE theory. For example, because an ODE behaves more rigidly, it is easier to study the representation capacities of NODES (input -> outputs representable mappings). For instance, [1] provides a sufficient condition for a NODE to be a Universal Approximator, [2] study the universal interpolation property of NODEs, and [3] establishes the existence of a large set of diffeomorphisms a NODE can represent.
> >
> > Also, a practical advantage of Neural ODEs is the continuous adjoint method: the ode can be integrated backward in time at the same time as gradients are computed so that the forward trajectory does not have to be stored, hence implying memory gains.
> >
> > [1] Li, Q., Lin, T., & Shen, Z. (2022). Deep learning via dynamical systems: An approximation perspective. Journal of the European Mathematical Society. https://arxiv.org/abs/1912.10382
> >
> > [2] Cuchiero, C., Larsson, M., & Teichmann, J. (2020). Deep neural networks, generic universal interpolation, and controlled ODEs. SIAM Journal on Mathematics of Data Science, 2(3), 901-919. https://arxiv.org/abs/1908.07838
> >
> > [3] Teshima, T., Tojo, K., Ikeda, M., Ishikawa, I., & Oono, K. (2020). Universal approximation property of neural ordinary differential equations. arXiv preprint arXiv:2012.02414 https://arxiv.org/abs/2012.02414
> >
> > We strongly hope to have convinced the reviewer that both our theoretical and experimental settings were of interest and that they will raise their score.

---

### Author Response · Authors · 2022-08-02
**General response to all the reviewers**

We first want to thank the reviewers for their reviews, their positive feedback on our work as well as for their remarks, advice and criticisms.

We are glad that reviewer rbYF found our paper “well written”, with a “solid” theoretical contribution. Regarding reviewer rbYF’s criticism/questions, in particular concerning the choice of our initialization scaling and on the usefulness of having a reduced memory cost for storing the activation, please refer to our response to their review. Indeed, in contrast to reviewer rbYF’s claims, we argue (with cited literature) that the weight scaling for very deep ResNets must depend on depth (of the form $\frac1{N^\beta}$ where $\beta$ controls the smoothness of the trajectory), and we provide evidence that storing the activations is really the memory bottleneck in practical applications. We clarified these points in the updated version of the paper.

We are pleased to see that reviewer 9X5i found our proposed discrete adjoint method “interesting”. We address their question in our response to their review.

Finally, we are heartened by the numerous positive comments on our paper by reviewer hARH. We thank the reviewer for all the details provided in their review. We took into account their comments on the updated version of the paper.

We took into account most of the remarks and advice made by the reviewers in an updated version of the paper, where modifications appear in red.

---

### Meta-Review · Area_Chair_RpzV · 2022-08-25

**Recommendation:** Accept
**Confidence:** Certain

**Metareview:**

The paper provided an error bound for the approximation between the hidden trajectory of ResNet and its Neural ODE. The trajectory value of ResNet is different for each layer, which is more close to the practice. Authors find a case (linear residual functions, small initial loss) where the error bound can converge to zero. By leveraging the connection in the special case, an adjoin method is designed  to estimate the gradient with a memory-free manner. This work can push our understanding about the connection between deep NN models (i.e., ResNet) and Neural ODE. The designed algorithm also called for our attention that, the connection can facilitate "memory-free" gradient estimation.

**Award:**

No

---

### Decision · Program_Chairs · 2022-09-14

Accept